# Unlocking the Capabilities of Large Vision-Language Models for Generalizable and Explainable Deepfake Detection

Peipeng Yu [1]   Jianwei Fei [2]   Hui Gao [1]   Xuan Feng [1]   Zhihua Xia [1]   Chip-Hong Chang [3]

## Abstract

Current Large Vision-Language Models (LVLMs) have demonstrated remarkable capabilities in understanding multimodal data, but their potential remains underexplored for deepfake detection due to the misalignment of their knowledge and forensics patterns. To this end, we present a novel framework that unlocks LVLMs' potential capabilities for deepfake detection. Our framework includes a Knowledge-guided Forgery Detector (KFD), a Forgery Prompt Learner (FPL), and a Large Language Model (LLM). The KFD is used to calculate correlations between image features and pristine/deepfake image description embeddings, enabling forgery classification and localization. The outputs of the KFD are subsequently processed by the Forgery Prompt Learner to construct fine-grained forgery prompt embeddings. These embeddings, along with visual and question prompt embeddings, are fed into the LLM to generate textual detection responses. Extensive experiments on multiple benchmarks, including FF++, CDF2, DFD, DFDCP, DFDC, and DF40, demonstrate that our scheme surpasses state-of-the-art methods in generalization performance, while also supporting multi-turn dialogue capabilities. Our code is available at `https://github.com/botianzhe/LVLM-DFD`.

## 1. Introduction

The rapid advancement of generative artificial intelligence has significantly accelerated the development of deepfake technology, facilitating realistic facial manipulation and reenactment. While these technologies have notable applications in the entertainment and art fields, such as Stable Diffusion (Esser et al., 2024) and DALL·E (Ramesh et al., 2021), their misuse poses critical security risks to society (Wang et al., 2024b). These tools allow users to synthesize realistic but nonexistent content by merely inputting carefully designed prompts, making deepfake generation more accessible and potentially dangerous than ever before.

Large Vision-Language Models (LVLMs) present a promising leverage to this issue. Pre-trained on extensive and diverse datasets, LVLMs capture vast amounts of knowledge about natural objects, providing significant potential to improve generalization in identifying manipulated content. An LVLM typically uses an image encoder to extract image features, which are then combined with text prompts and input into the Large Language Model (LLM) to generate responses. For instance, inputting an image along with a prompt like "*This is a facial image designed for deepfake detection, and it should not exhibit any localized color discrepancies or evident signs of splicing. Is this a deepfake image?*" allows the LVLM to assess potential manipulations. However, existing LVLMs are primarily optimized for general image understanding tasks and may not effectively capture the fine-grained features required for deepfake detection. Directly performing fine-tuning presents challenges, as the LVLM may struggle to interpret specialized terms like "*color discrepancies*" or "*visual artifacts*" as intended in the context of forgery detection. Therefore, it is crucial to design fine-grained prompt embeddings to facilitate LVLM training.

In this paper, we aim to unlock the capabilities of Large Vision-Language Models for deepfake detection tasks. Humans naturally use specific descriptors, such as *subtle visual artifacts*, *localized lighting inconsistencies*, and *overly smoothed textures*, to characterize manipulated content. However, these features are difficult to accurately replicate by data simulation or feature augmentation alone, limiting the ability of current methods to fully interpret manipulated content (Zhang et al., 2024). To address this limitation, we propose to explore the correction between images and textual descriptions to assist deepfake detection. As shown in Figure 1, we propose a novel LVLM-based deepfake detection framework under the guidance of pretrained knowledge

[1] College of Cyber Security, Jinan University [2] University of Macau [3] School of Electrical and Electronic Engineering, Nanyang Technology University. Correspondence to: Zhihua Xia <xia_zhihua@163.com>.

*Proceedings of the 42nd International Conference on Machine Learning*, Vancouver, Canada. PMLR 267, 2025. Copyright 2025 by the author(s).

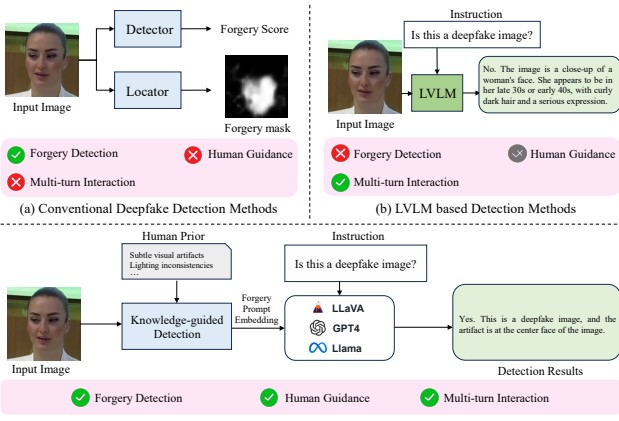

*Figure 1.* **Comparison of existing deepfake detection methods.** Existing approaches can perform localization and classification but ignore integrating external knowledge. Our scheme introduces a Knowledge-guided detection module to enhance generalization capabilities, and enables multi-turn dialogues.

to perform generalizable and explainable detection (Lester et al., 2021; Liu et al., 2022). We first integrate external knowledge to train a forgery detector, and then incorporate its features into LLM to generate responses. Specifically, the framework comprises two key stages: Knowledge-guided Forgery Detector Training and LLM Prompt Tuning. In the Knowledge-guided Forgery Detector Training phase, we aim to train a high-precision deepfake detector. Leveraging pre-trained multimodal encoders, we extract image features from images and textual features from descriptions of pristine and forged images. By calculating the correlation between these image features and description text embeddings, we generate consistency maps that represent the alignment between visual content and textual descriptions. These maps are subsequently processed by the Forgery Locator and Classifier to produce a forgery segmentation map and forgery score. Subsequently, in the LLM Prompt Tuning phase, we incorporate deepfake detection knowledge into the LLM to generate forgery detection descriptions. To ensure accurate deepfake detection, we train the LVLM using simulated forgery image-text pairs specifically tailored for this task. Our contributions are summarized as follows:

- We propose a novel LVLM-based framework for deepfake detection that integrates fine-grained forgery prompt embeddings through prompt tuning, which significantly enhances model generalization and explainability.

- We introduce a Knowledge-guided Forgery Detector to generate forgery consistency maps to align image features with description text embeddings of both pristine and deepfake images for enhanced generalization.

- Extensive experiments on multiple benchmarks, including FF++, CDF1, CDF2, DFD, DFDCP, DFDC, and DF40, demonstrate that our scheme outperforms existing methods in generalization performance, with the added capability of supporting multi-turn dialogue.

## 2. Related Works

**Deepfake Detection Methods:** Conventional classification architectures have achieved significant success in detecting forgery clues in early deepfake content. Various strategies, such as data augmentation (Li et al., 2020a; Shiohara & Yamasaki, 2022; Nguyen et al., 2024), feature consistency analysis (Zhao et al., 2021b; Yan et al., 2023), and frequency domain anomaly detection (Jeong et al., 2022; Liu et al., 2021; Wang et al., 2023a), have been explored in recent years to enhance the generalization of deepfake detection models. While these methods achieve a high detection accuracy, they primarily rely on data augmentation or enhanced feature extraction (Yan et al., 2024), and often neglect the integration of external human knowledge. However, many deepfake characteristics are embedded in human knowledge, which is challenging to capture through data or feature augmentation alone. This limitation significantly constrains the generalization capabilities of existing algorithms. In this paper, we propose an LVLM-based deepfake detection framework that aligns image features with real/fake descriptions to enhance the model's capacity to detect unseen deepfakes.

**Large Vision-Language Models:** Recent advancements in Large Vision-Language Models (LVLMs) have showcased their potential in multimodal tasks (Gunjal et al., 2024; Leng et al., 2024; Gu et al., 2024). A typical LVLM architecture comprises an image encoder, a projector, and a LLM. The image encoder extracts visual features from input images, which are then transformed by the projector into visual prompt embeddings. These visual embeddings, combined with textual prompt embeddings, are fed into the LLM to generate responses. Building on this architecture, models such as BLIP-2 (Li et al., 2023), LLaVA (Liu et al., 2024a), and MiniGPT-4 (Zhu et al., 2024) have achieved notable advancements in language instruction following (Su et al., 2023; Yang et al., 2024) and visual reasoning (Chen et al., 2024) for natural scenes. Some studies have also explored the application of LVLMs in forgery detection. FakeShield (Xu et al., 2024) constructed a large-scale image-text dataset and introduced an LVLM-based framework specifically designed for forgery detection. FKA-Owl (Liu et al., 2024b) proposed a novel fake news detection framework that leverages forgery-specific knowledge to augment LVLMs, enabling them to reason about manipulations. Similarly, FFAA (Huang et al., 2024) proposed a multimodal LVLM approach for explainable, open-world face forgery analysis, highlighting the potential of LVLMs in

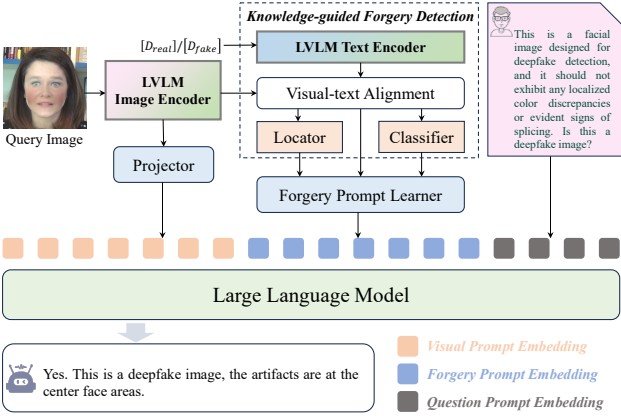

*Figure 2.* **Overview of the proposed framework.** The framework consists of three main components: (1) a Knowledge-guided Forgery Detection module, structured as a multi-task learning framework comprising three parts: a Visual-text alignment module, a forgery classifier, and a forgery locator. (2) a Forgery Prompt Learner, which is designed to fuse forgery detection outputs and generate forgery prompt embeddings. (3) a Large Language Model (LLM), which utilizes the extracted question, visual, and forgery prompt embeddings to generate responses.

forgery detection tasks.

Despite these advancements, current LVLMs primarily focus on general language processing and visual understanding, often missing the fine-grained details that are essential for deepfake detection tasks. This limitation restricts their effectiveness in forgery localization and classification. To bridge this gap, we develop a novel deepfake detection framework based on LVLMs that constructs fine-grained forgery prompt embeddings to guide the LLM in detecting subtle manipulations. By integrating rich external knowledge in the pretrained LVLM, our scheme could enhance generalization across diverse forgery types while retaining the model's original dialogue capabilities.

## 3. Proposed Method

Our objective is to enable LVLMs to accurately distinguish between real and fake faces. Although LVLMs are trained on large-scale datasets, they are primarily trained for general image understanding tasks and often lack the sensitivity for detecting forgery details. To address this limitation, we propose a novel LVLM-based deepfake detection framework that enhances sensitivity to deepfake artifacts through constructing fine-grained forgery prompts. As shown in Figure 2, our scheme builds upon a conventional LVLM framework, which comprises an image encoder, a projector, and an LLM. The image encoder extracts content features from input images, which are subsequently transformed into visual prompt embeddings $E_{visual}$ by the projector.

Additionally, user queries are encoded as question prompt embeddings $E_{question}$. To train the model for forgery detection, we employ a two-stage process. In the first stage, we train a Knowledge-guided Forgery Detector (KFD) to perform forgery detection and localization by calculating the correlation between image content features and descriptions of pristine/deepfake images. This stage ensures that the KFD can effectively classify and localize forgery artifacts by learning fine-grained visual-text correlations. In the second stage, we perform LLM Prompt Tuning to integrate the KFD knowledge into the LVLM framework. Specifically, we design a Forgery Prompt Learner to convert forgery-related features into forgery prompt embeddings. These embeddings, along with the question and visual prompt embeddings, are then fed into the LLM to generate a textual detection result. To further enhance the interpretability of our model, we adopt an alternating training strategy using both deepfake datasets and general Visual Question Answering (VQA) datasets. This enables the model to accurately detect deepfakes while retaining multi-turn dialogue capabilities.

### 3.1. Knowledge-guided Forgery Detector

**Forgery Visual-text Alignment:** To acquire forgery detection-related knowledge, inspired by (Jeong et al., 2023), we align image content features with predefined text description features to obtain fine-grained forgery features. This process is illustrated in Figure 3. Specifically, this process involves a pretrained image encoder and a pretrained text encoder. Both the image and text encoders are sourced from ImageBind (Girdhar et al., 2023), a large-scale multimodal pre-trained model with extensive cross-modal knowledge. We first define real and fake image descriptions ($D_{real}$ and $D_{fake}$) and utilize a text encoder to extract their semantic features. These features are concatenated with a learnable embedding to obtain the task-specific text embedding $F_{text} \in \mathbb{R}^{2 \times C_{text}}$, where $C_{text}$ denotes the dimensionality of the text embedding. For the visual features, we select $l$ layers from the image encoder and obtain the intermediate features extracted by each selected layer. The extracted features are then processed by linear layers to generate visual features $F_{vis}^i \in \mathbb{R}^{H_i \times W_i \times C_{text}}$, where $i$ indicates the $i$-th layer. The similarity maps between visual features and textual features are calculated and concatenated as consistency maps. The formula for computing the consistency maps is as follows:

$$\rho_{text} = \{F_{vis}^i F_{text}^\top\}. \tag{1}$$

Additionally, to optimize the extracted image features, we compute the cosine similarity between the features of a reference image (pristine image) $F_{vref}^i$ and the input image features $F_{vis}^i$. This similarity optimization enhances the robustness of the features extracted by the image encoder.

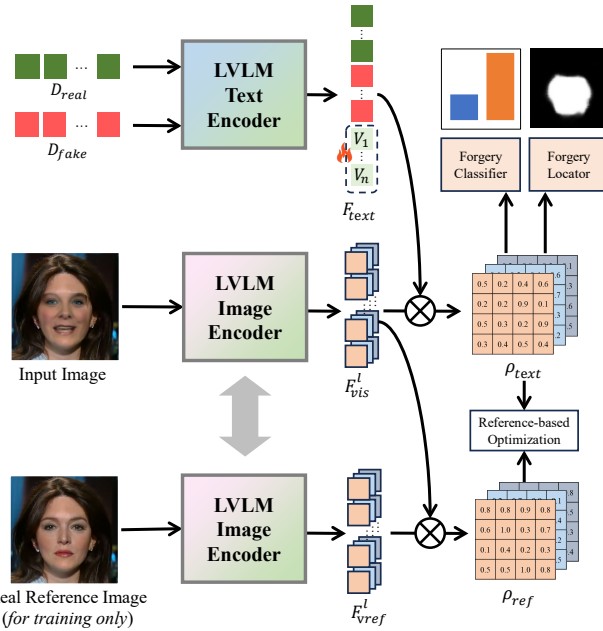

*Figure 3.* **Overview of the Knowledge-guided Forgery Detector.** It computes correlation between image and text modalities to assist forgery classification and localization. The Reference based Optimization process is applied exclusively during the training phase to enhance the robustness of extracted features.

Note that reference images are used only during training and are not employed at inference time. The similarity is calculated as:

$$\rho_{ref} = \{Cos(F_{vref}^i, F_{vis}^i)\}. \tag{2}$$

**Forgery Locator and Classifier:** To enhance the model's sensitivity to deepfake content, we introduce a Forgery Locator and a Forgery Classifier to locate forgery areas and classify pristine and deepfake images. The Forgery Locator consists of three branches. Each branch performs downsampling and up-sampling operations on the corresponding consistency maps, followed by interpolation, concatenation, and convolutional layers to generate the segmentation map. The Forgery Classifier also consists of three branches. The three feature maps are first processed by convolution and pooling operations, and then concatenated to form a unified feature representation. After that, we use two fully connected layers to calculate the probability of the image being real or fake. Here, we use Dice Loss to improve the accuracy of forgery segmentation. We further enhance the robustness of extracted forgery features by optimizing the matching degree between the textual consistency map ($\rho_{text}$) and the reference consistency map ($\rho_{ref}$). Both maps are expected to accurately localize the forged regions. The localization loss is formulated as follows:

$$\mathcal{L}_{loc} = Dice(\phi(\rho_{text}), gt) + \lambda Dice(\phi(\rho_{ref}), gt), \tag{3}$$

where $\phi$ is the Forgery Locator, and $gt$ is the ground truth mask. Dice loss optimizes the overlap between the predicted segmentation and the ground truth. $\lambda$ is the loss weight for balancing these two losses.

Additionally, we use binary cross-entropy loss to optimize the performance of the forgery classification task. The formula is as follows:

$$\mathcal{L}_{cls} = -\left(c\log(\hat{c}) + (1 - c)\log(1 - \hat{c})\right), \tag{4}$$

where $\hat{c}$ is the predicted forgery score that the image is fake, and $c$ is the ground truth label (0 for real and 1 for fake).

### 3.2. Forgery Prompt Learner and LLM

**Forgery Prompt Learner:** To effectively convert forgery-related features into inputs for the LLM, we propose a Forgery Prompt Learner to transform the forgery segmentation map, the forgery score, and the consistency maps into forgery prompt embeddings. At the same time, we add a learnable prompt embeddings for the forgery prompt learner to incorporate extra information for the deepfake detection task. The Forgery Prompt Learner consists of two convolutional neural networks, one fully connected layer, and the learnable prompt embeddings $E_{base} \in \mathbb{R}^{n_1 \times C_{emb}}$, where $C_{emb}$ represents the dimensionality of the embedding vectors. Specifically, the two convolutional networks transform the forgery segmentation map and consistency maps into vector representations, $E_{loc} \in \mathbb{R}^{n_2 \times C_{emb}}$ and $E_{cons} \in \mathbb{R}^{n_3 \times C_{emb}}$, respectively. The forgery score is expanded into $E_{cls} \in \mathbb{R}^{1 \times C_{emb}}$. These embeddings are concatenated and fed into the convolution layer to generate the forgery prompt embeddings $E_{forgery} \in \mathbb{R}^{n_f \times C_{emb}}$. Finally, the forgery prompt embeddings, visual prompt embeddings, and question prompt embeddings are input into the LLM.

**LLM:** The LLM processes the prompt embeddings to interpret the context and accurately identify forged regions. By integrating visual details (from $E_{forgery}$ and $E_{visual}$) with user queries (from $E_{question}$), the LLM produces responses that provide forgery detection judgments and precisely localize manipulated regions (e.g., eyes, mouth). Here, we employ prompt tuning and LoRA to fine-tune the LLM using simulated image-text pairs specifically tailored for deepfake detection tasks. To ensure the LLM generates accurate responses, we use cross-entropy loss to quantify the discrepancy between the predicted response and the target label. The formula is defined as follows:

$$\mathcal{L}_{llm} = -\sum_j y_j \log(\hat{y}_j), \tag{5}$$

where $\hat{y}_j$ represents the predicted probability for the $j$-th token, and $y_j$ denotes the corresponding ground truth label.

### 3.3. Data for LLM Prompt Tuning

**Forgery Data Simulation:** We intend for the LLM to identify pristine and deepfake images, while also locating the forged regions. This requires training on image-text pairs specifically depicting manipulated areas, which are currently unavailable. To address this gap, we draw on techniques from SBI (Shiohara & Yamasaki, 2022) to construct image-text pairs using existing real images. First, we generate facial landmarks from a real image $I_{real}$, then randomly select 1 to $n$ regions (e.g., the nose, mouth, or eyes) as target forgery areas. We apply a slight affine transformation to the real image, resulting in an affine-transformed image $I_{affine}$. The original real image is used as the background (target face), and the affine-transformed image serves as the foreground (source face). Following the approach in (Nguyen et al., 2024), we apply Poisson blending to combine the foreground and background images. The blending process is defined as follows:

$$\mathbf{I}_M = \mathbf{M} \odot \mathbf{I}_{affine} + (1 - \mathbf{M}) \odot \mathbf{I}_{real} , \qquad (6)$$

where $\mathbf{M}$ is the Convex Hull mask constructed based on the selected forgery region, with values ranging from 0 to 1. The symbol $\odot$ denotes element-wise multiplication.

**Question and Answer Content:** Training an LVLM requires a large number of visual question-answer pairs. Therefore, we construct corresponding text queries for each image. To ensure compatibility with the deepfake detection task, we first include a background description in each query, for example: "*This is a facial image designed for deepfake detection, and it should not exhibit any localized color discrepancies or evident signs of splicing.*", which can be regarded as a kind of human prior knowledge. Additionally, we incorporate the prediction results from the Knowledge-guided Forgery Detector (KFD) into the prompt, such as "*According to KFD prediction, the forgery score is 0.93.*" We then ask a question related to the content of the image, such as "*Is this a deepfake image?*" The LVLM's response states whether a forgery is present in the image and where the forgery area is. For instance, "*Yes. This is a deepfake image, and the artifact is at the center face of the image.*" Here, the forgery regions are defined according to those selected during forgery data simulation. By defining both queries and responses, we can train the LVLM to distinguish between pristine and deepfake images. The prompt format input to the LVLM follows this format:

```
###Human: E_visual</Img>E_forgery[Task
description][KFD result] Is this a
deepfake image?  ###Assistant:,
```

where $E_{visual}$ represents visual prompt embeddings, $E_{forgery}$ denotes the forgery prompt embeddings learned by the Forgery Prompt Learner, `KFD result` indicates the forgery classification results, and `Task description`

provides a textual description of deepfake detection task.

## 4. Experiments

### 4.1. Experimental Settings

**Datasets.** The FaceForensics++ (FF++) (Rossler et al., 2019) dataset includes 1,000 real videos and 5,000 forgery videos across five deepfake categories, which is one of the most widely-used datasets for deepfake detection. DFD (Dufour & Gully, 2019), CDF1, CDF2 (Li et al., 2020b), DFDCP (Dolhansky, 2019), DFDC (Dolhansky et al., 2020), and DF40 (Yan et al.) are commonly used datasets for evaluating generalization performance in deepfake detection. The images are all cropped using Dlib and RetinaFace. We train only on real data from the FF++ dataset.

**Evaluation Metrics.** Following existing approaches (Shiohara & Yamasaki, 2022; Nguyen et al., 2024), we use the video-level Area Under the Receiver Operating Characteristic Curve (AUC) and Average Precision (AP) as our evaluation metric. Additionally, we assess the LLM's performance by evaluating the binary classification (Yes or No) of authenticity in its textual output, allowing us to calculate a corresponding video-level AUC for the LLM's responses.

**Compared Methods.** We evaluate our approach against several state-of-the-art deepfake detection algorithms (Rossler et al., 2019; Li et al., 2020a; Qian et al., 2020; Zhao et al., 2021a; Liu et al., 2021; Zhao et al., 2021b; Cao et al., 2022; Shiohara & Yamasaki, 2022; Wang et al., 2023a;b; Dong et al., 2023; Yan et al., 2023; Xu et al., 2023; Yan et al., 2024; Nguyen et al., 2024; Cheng et al., 2024; Lin et al., 2025; Luo et al., 2024; Ba et al., 2024; Fu et al., 2025) and LVLM-based Methods (Khan & Dang-Nguyen, 2024; Su et al., 2023; Liu et al., 2024b; Wang et al., 2024a)

**Implementation Details.** Our approach leverages the PandaGPT architecture, which incorporates the ImageBind-Huge model as the image and text encoder. We extract features from the 16th, 24th, and 32nd layers of the encoder to compute consistency maps with the text features, which are then passed to the Vicuna-7B model for inference. For multi-turn dialogue capability, we alternate training between the deepfake dataset and the PandaGPT dataset. The forgery region number $n$ is set as 3. All the images are cropped to $224 \times 224$. Training is conducted on two Nvidia RTX 4090 GPUs over 50 epochs, using the Adam optimizer with a learning rate of 1e-4 and a weight decay of 1e-5. The loss parameter $\lambda$ is set as 1.

### 4.2. Comparison with SOTA Detection Methods

We first compare our approach with several state-of-the-art deepfake detection methods (Li et al., 2020a; Shiohara & Yamasaki, 2022; Cao et al., 2022; Huang et al., 2023; Yan

| Methods | | Intra-dataset | | Cross-dataset | | | | | | | | | |
|---|---|---|---|---|---|---|---|---|---|---|---|---|---|
| | | FF++ | | CDF2 | | DFD | | DFDC | | CDF1 | | DFDCP | |
| | | AUC | AP | AUC | AP | AUC | AP | AUC | AP | AUC | AP | AUC | AP |
| Xception (Rossler et al., 2019) | CVPR'19 | 97.23 | 96.44 | 81.65 | 88.74 | 89.75 | - | - | - | 80.98 | 90.00 | 69.90 | 81.95 |
| FaceXRay+BI (Li et al., 2020a) | CVPR'20 | - | - | 79.50 | - | 95.40 | 93.34 | - | - | 80.58 | 73.33 | 80.92 | 72.65 |
| F3Net (Qian et al., 2020) | ECCV'20 | 98.20 | 96.17 | 78.88 | 86.23 | 93.33 | 97.70 | 71.77 | 71.99 | 81.11 | 89.61 | 73.50 | 79.30 |
| Multiattentional (Zhao et al., 2021a) | CVPR'21 | - | - | 68.26 | - | 92.95 | - | - | - | - | - | 63.02 | - |
| SPSL (Liu et al., 2021) | CVPR'21 | 96.91 | 89.37 | 79.86 | 87.83 | 96.12 | 98.20 | 66.16 | 71.13 | 85.02 | 92.17 | 75.86 | 82.64 |
| PCL+I2G (Zhao et al., 2021b) | CVPR'21 | 99.11 | - | 90.03 | - | 99.07 | - | - | - | - | - | 74.27 | - |
| RECCE (Cao et al., 2022) | CVPR'22 | 99.32 | 97.25 | 82.31 | 88.26 | 98.26 | 98.40 | 69.58 | 69.98 | 81.49 | 89.04 | 71.49 | 77.04 |
| SBI (Shiohara & Yamasaki, 2022) | CVPR'22 | 99.15 | 99.15 | 93.82 | 92.99 | 96.32 | 96.13 | 74.47 | 74.09 | 93.44 | 93.77 | 90.95 | 85.98 |
| SFDG (Wang et al., 2023a) | CVPR'23 | - | - | 75.83 | - | 88.00 | - | - | - | - | - | 73.63 | - |
| AltFreezing (Wang et al., 2023b) | CVPR'23 | 93.81 | 98.74 | 89.50 | 87.41 | 98.50 | 94.59 | 64.75 | 67.52 | 88.48 | 92.79 | 64.05 | 76.22 |
| CADDM (Dong et al., 2023) | CVPR'23 | 99.70 | - | 93.88 | - | 99.03 | - | 73.85 | - | 85.68 | - | 74.19 | - |
| UCF (Yan et al., 2023) | CVPR'23 | 98.69 | 97.99 | 83.73 | 90.10 | 94.50 | 98.04 | 75.11 | 74.76 | 86.08 | 91.78 | 80.50 | 77.16 |
| TALL (Xu et al., 2023) | ICCV'23 | 99.87 | - | 90.79 | - | - | - | 76.78 | - | - | - | - | - |
| LSDA (Yan et al., 2024) | CVPR'24 | - | - | 91.10 | - | - | - | 77.00 | - | - | - | - | - |
| LAANet (Nguyen et al., 2024) | CVPR'24 | 99.96 | - | 95.40 | 97.64 | 99.51 | 99.40 | - | - | - | - | 86.94 | 97.70 |
| CFM (Luo et al., 2024) | TIFS'24 | - | - | 89.65 | - | - | - | 80.22 | - | - | - | - | - |
| ED (Ba et al., 2024) | AAAI'24 | - | - | 93.6 | - | - | - | 75.4 | - | - | - | 90.2 | - |
| ProDet (Cheng et al., 2024) | NIPS'24 | - | - | 92.62 | 96.05 | - | - | 71.52 | 72.8 | 94.48 | 96.66 | 82.83 | 88.89 |
| RepDFD (Lin et al., 2025) | AAAI'25 | - | - | 89.94 | - | - | - | 80.99 | - | - | - | 95.03 | - |
| UDD (Fu et al., 2025) | AAAI'25 | - | - | 93.13 | - | - | - | 81.21 | - | - | - | 88.11 | - |
| Ours | | 99.53 | 99.54 | 94.71 | 93.59 | 99.64 | 99.68 | 79.12 | 77.69 | 97.62 | 97.67 | 91.81 | 88.26 |

*Table 1.* **Intra-dataset and cross-dataset evaluation of KFD compared with existing deepfake detection methods**. The highest AUC and AP scores among all methods are highlighted in **bold**, while the second-highest scores are underlined. All experimental results are sourced from the original papers or reproduced using publicly available code repositories.

et al., 2024; Tan et al., 2024).

**Intra-Dataset Evaluation.** Following the intra-dataset protocol outlined in previous works (Yan et al., 2024; Nguyen et al., 2024), we compare our approach with existing state-of-the-art deepfake detection methods based on the outputs of KFD. As shown in Table 1, our method achieves competitive results, reaching a detection AUC of 99.53% on the FF++ dataset.

**Cross-Dataset Evaluation.** Following previous works (Li et al., 2020a; Shiohara & Yamasaki, 2022; Nguyen et al., 2024), we further perform a cross-dataset evaluation. The models are trained on real data from the FF++ dataset, and the detection performance is evaluated on CDF1, CDF2, DFD, DFDCP, and DFDC datasets. We report video-level AUC and AP scores across these datasets in Table 1. As summarized in Table 1, our method achieves superior video-level AUC and AP scores across these diverse datasets, outperforming state-of-the-art baselines.

**Cross-Manipulation Evaluation of KFD.** To evaluate the robustness of our algorithm against unknown forgery techniques, we assess its detection performance on data generated by unseen manipulation methods within the DF40 dataset. DF40 comprises synthetic deepfake samples generated from real images in the FF++ dataset, encompass-

| | | UniFace | InSwapper | FSGAN | FaceDancer | e4s |
|---|---|---|---|---|---|---|
| **DF40-FS** | SBI | 89.02 | **88.52** | 89.62 | 78.18 | 86.36 |
| | CADDM | 86.86 | 78.65 | 88.86 | 76.54 | 87.92 |
| | Ours | **90.61** | 87.64 | **93.75** | **82.97** | **94.68** |
| | | PIRender | HyperReenact | FOMM | FS_vid2vid | LIA |
| **DF40-FR** | SBI | 81.81 | 65.31 | 88.05 | **83.72** | 89.22 |
| | CADDM | 77.37 | 69.26 | 84.77 | 72.86 | 69.67 |
| | Ours | **88.29** | **81.55** | **93.34** | 71.56 | **99.99** |
| | | StyleGAN2 | StyleGAN-XL | DDIM | DiT-XL/2 | PixArt-$\alpha$ |
| **DF40-EFS** | SBI | 97.91 | 23.26 | 99.56 | 79.04 | 98.78 |
| | CADDM | **100.00** | 98.69 | 98.10 | 79.90 | 99.74 |
| | Ours | **100.00** | **100.00** | **99.93** | **94.18** | **100.00** |

*Table 2.* **Cross-manipulation evaluation on DF40 dataset.** All methods are evaluated on various types of manipulated subsets of DF40. The best-performing values are highlighted in bold.

ing a diverse range of manipulation methods. Our models are trained exclusively on real images from FF++ and subsequently tested on unseen manipulation types within DF40. As presented in Table 2, our approach consistently outperforms existing methods across multiple forgery types, achieving state-of-the-art detection results. Notably, our method demonstrates robust generalization capabilities when confronted with advanced deepfake generation techniques, including FSGAN, E4S, LIA, StyleGAN, DDIM,

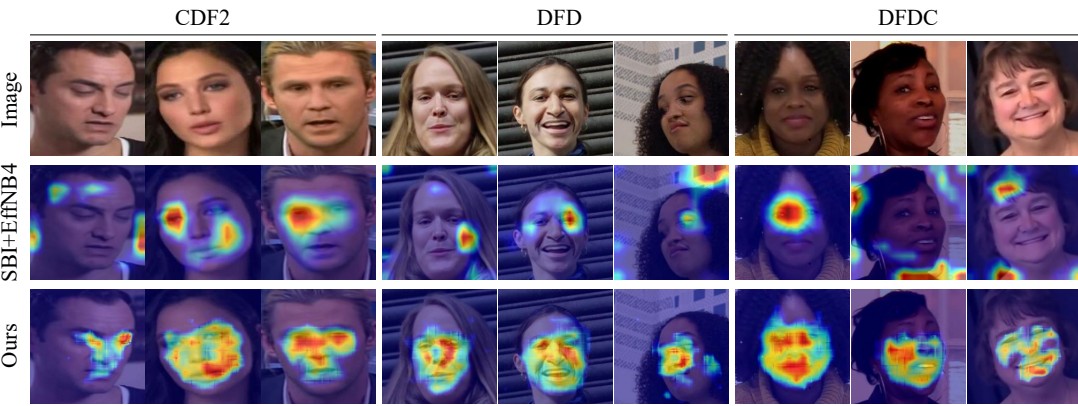

*Figure 4.* GradCAM visualizations on the CDF2, DFD, and DFDC datasets.

| Methods | FF++ | CDF2 | DFD | CDF1 | DFDCP | Avg |
|---------|------|------|-----|------|-------|-----|
| Clipping | 91.32 | 79.97 | 88.74 | 81.45 | 69.26 | 82.15 |
| RepDFD | - | 89.94 | - | - | **95.03** | 92.49 |
| UDD | - | 93.13 | - | - | 88.11 | 90.62 |
| Ours-KFD | **99.53** | **94.71** | **99.64** | **97.62** | 91.81 | **96.66** |
| PandaGPT | 63.42 | 62.53 | 64.56 | 55.01 | 46.06 | 58.32 |
| Qwen2-VL | 50.50 | 48.88 | 42.38 | 46.46 | 51.71 | 47.99 |
| FAK-Owl | 67.54 | 60.85 | 67.99 | 69.84 | 70.72 | 67.39 |
| Ours-LVLM | **97.11** | **89.90** | **92.26** | **95.97** | **86.70** | **92.39** |

*Table 3.* **Comparison of our scheme with existing LVLM-based methods.** Clipping (Khan & Dang-Nguyen, 2024) and PandaGPT (Su et al., 2023) are fine-tuned on the FF++ dataset, FAK-Owl (Liu et al., 2024b) is trained based on original image-text pairs, and Qwen2-VL (Wang et al., 2024a) is evaluated with its original pre-trained weights. The AUC scores for Ours-LVLM are derived from the textual output of the LLM, and the AUC scores for Ours-KFD are derived from the KFD output. The best and second-best results are highlighted in **bold** and underlined, respectively.

and PixArt-$\alpha$.

**GradCAM Visualization.** To further interpret the decision process of our model, we use GradCAM (Selvaraju et al., 2017) to visualize attention when encountering unknown datasets. As shown in Figure 4, we compare our method with SBI (Shiohara & Yamasaki, 2022) across the CDF2, DFD, and DFDC datasets. While SBI can detect forged regions, it may misidentify these areas and tends to highlight irrelevant regions when encountering unknown forgeries. In contrast, our scheme, guided by external knowledge, effectively identifies forgery regions.

### 4.3. Comparison with LVLM-based Methods

**Detection Performance of LVLM.** We benchmark our framework against state-of-the-art LVLM-based classification methods (Khan & Dang-Nguyen, 2024; Lin et al., 2025; Fu et al., 2025) and VQA methods (Su et al., 2023; Wang

et al., 2024a; Liu et al., 2024b). In these evaluations, both the image and the corresponding query are provided as inputs to the LVLM, and the model is required to determine the authenticity of the image (real or fake). For classification task, our Knowledge-guided Forgery Detector (KFD) demonstrates significantly superior detection performance compared to existing LVLM-based classification methods. For VQA models such as PandaGPT, Qwen2-VL, and FAK-Owl, we utilize the LLM's output ("Yes" or "No") to classify authenticity and subsequently compute the AUC values for evaluation. As summarized in Table 3, our method consistently outperforms existing LVLM-based VQA methods, achieving better performance on the FF++, CDF2, DFD, CDF1, and DFDCP datasets.

**Dialogues Visualization.** Unlike prior detection methods, our approach not only supports deepfake detection but also facilitates multi-turn dialogues, enabling users to further inquire about the image content. Figure 5 presents some example dialogues under intra-dataset and cross-dataset evaluations. It shows that our proposed scheme accurately identifies forged regions within the images, while the LLM provides precise and contextually relevant judgments. Additional multi-turn dialogue examples are provided in the supplementary material.

### 4.4. Analysis

**Number of Training Images.** Obtaining large-scale face images in real-world scenarios is often infeasible. Hence, we assess our model's performance with varying numbers of training images. Specifically, we randomly sample 50, 100, 200, and 500 real images from FF++ to create corresponding fake image-text pairs for training. The training steps are fixed as 500. The video-level AUC is calculated based on the responses of LLM. As shown in Table 4, our approach achieves state-of-the-art performance using only 500 training images. Although reducing the training set size

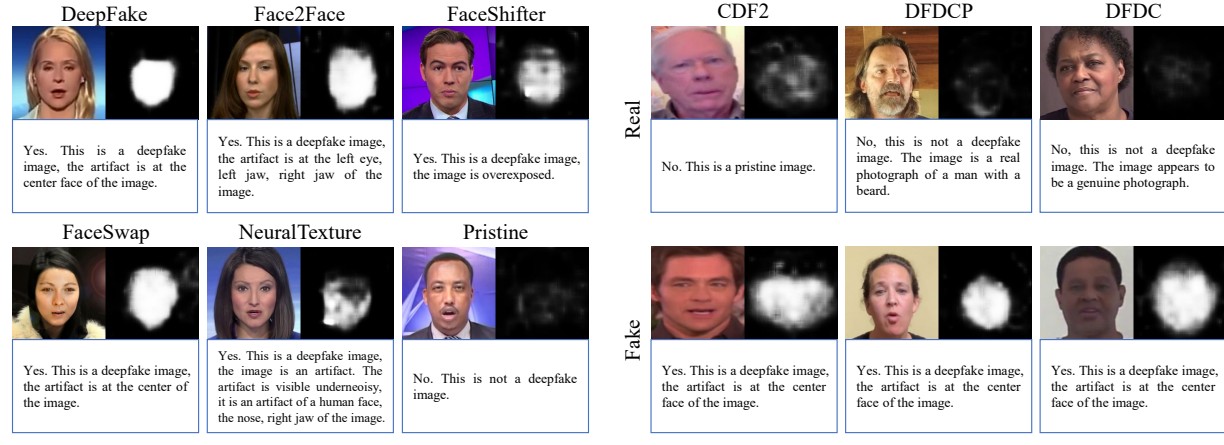

(a) Intra-dataset Evaluations                    (b) Cross-dataset Evaluations

*Figure 5.* **Forgery Localization and LLM Responses under Intra-dataset and Cross-dataset Evaluations.** Each example includes the original image (top-left), the corresponding forgery segmentation map (top-right), and the textual detection result generated by the LLM (bottom).

| Number | Test set AUC | | | | |
|---|---|---|---|---|---|
| | CDF1 | CDF2 | DFD | DFDC | DFDCP |
| 50 | 77.78 | 82.64 | 82.25 | 70.49 | 84.70 |
| 100 | 82.59 | 81.74 | 86.34 | 72.72 | **85.06** |
| 200 | **84.53** | 80.68 | 88.14 | 74.12 | 83.48 |
| 500 | 84.17 | **83.31** | **88.17** | **74.23** | 84.65 |

*Table 4.* **Generalization evaluation across different numbers of training images.**

| FPL | LLM | Lora | FF++ | | CDF2 | | DFDC | |
|---|---|---|---|---|---|---|---|---|
| | | | AUC | AP | AUC | AP | AUC | AP |
| | ✓ | | 50.00 | 50.00 | 50.28 | 65.56 | 50.00 | 49.93 |
| | ✓ | ✓ | 63.42 | 61.22 | 62.53 | 59.02 | 55.11 | 53.47 |
| ✓ | ✓ | | 96.59 | 96.13 | 89.25 | **94.69** | 67.65 | 64.07 |
| ✓ | ✓ | ✓ | **97.11** | **96.97** | **89.90** | 94.51 | **68.77** | **64.57** |

*Table 5.* **Ablation study results on FF++, CDF2, and DFDC.** The ✓ in the FPL column indicates the inclusion of forgery prompt embeddings in the framework. The ✓ in the LLM column signifies the use of an LLM for inference, while the ✓ in the LoRA column denotes whether LoRA is used to fine-tune the LLM.

| Training Set | | Test Set AUC (%) | | | | |
|---|---|---|---|---|---|---|
| Dataset | #Real | FF++ | CDF2 | DFDCP | DFDC | Avg |
| FF++ | 720 | 99.54 | 94.32 | 92.39 | 77.03 | 90.82 |
| CDF2 | 622 | 97.95 | 95.79 | 86.36 | 70.27 | 87.59 |
| DFDCP | 737 | 99.15 | 85.17 | 93.74 | 69.64 | 86.93 |

*Table 6.* **Generalization evaluation across various training datasets.**

leads to a slight degradation in accuracy, our method still maintains competitive results with as few as 100 training images. This underscores the robustness of our framework, particularly in scenarios where data are scarce.

**Effect of Prompt Tuning.** The prompt tuning process is designed to convert forgery detection knowledge into the input of LLM to facilitate accurate detection. This process involves the Forgery Prompt Learner (FPL), the LLM, and the LoRA strategy. To evaluate the effectiveness of this component, we conduct ablation experiments on FF++, CDF2, and DFDC datasets. For each configuration, the AUC is calculated based on the LLM's output in determining real versus fake images. As shown in Table 5, models equipped with the Forgery Prompt Learner demonstrate higher AUC values, indicating improved effectiveness for deepfake detection tasks. Furthermore, the integration of LoRA further enhances performance, achieving superior results across multiple datasets compared to configurations without LoRA.

**Generalizability to Training Datasets.** The generalizability of deepfake detection is closely related to the diversity of the training data used. To verify the effectiveness of

our approach across different datasets, we train the model on various training sets and conducted cross-dataset evaluation on FF++, CDF2, DFDCP, and DFDC datasets. We calculate detection AUC values based on the LLM's responses, as shown in Table 6. The results indicate that our method exhibits strong robustness across diverse datasets and demonstrates its ability to generalize across different types of forgeries using varied real data.

**Effects of different LLM architectures.** The deepfake detection performance is also related to the specific LLM architecture used, as different LLMs exhibit unique charac-

| LLM Architecture | Test set AUC (%) | | | | |
|---|---|---|---|---|---|
| | FF++ | CDF1 | DFD | DFDC | Avg |
| Llama-3.2-1B | 96.49 | 95.64 | 82.74 | 65.66 | 85.13 |
| Llama-3.2-3B | 97.08 | 95.30 | 82.80 | 66.12 | 85.32 |
| Vicuna-7B | **97.11** | **95.97** | **84.26** | **67.65** | **86.25** |

*Table 7.* **Generalization evaluation across different LLM architectures.** The AUC is computed based on the LLM output.

| Methods | Test set AUC (%) | | | | |
|---|---|---|---|---|---|
| | CDF1 | CDF2 | DFDC | DFDCP | Avg |
| SBI | 93.44 | 93.82 | 74.47 | 90.95 | 88.17 |
| w/o ROP | 96.18 | 92.77 | **79.31** | 91.73 | 90.00 |
| w/ ROP | **97.62** | **94.71** | 79.12 | **91.81** | **90.82** |

*Table 8.* **Ablation study on the Reference-based Optimization Process.** The highest AUC scores among all methods are highlighted in **bold**.

teristics. To examine detection performance across various LLM architectures, we evaluate three models: Llama-3.2-1B, Llama-3.2-3B, and Vicuna-7B. The evaluation is conducted on the FF++, CDF1, DFD, and DFDC datasets. As shown in Table 7, we observe that larger-scale LLM architectures consistently yield superior detection performance. This trend suggests that models with increased parameter capacity are more adept at capturing fine-grained forgery artifacts.

**Ablation Study on the Reference-based Optimization.** To validate the effectiveness of the reference based optimization process, we train the model under two configurations: with and without Reference-based Optimization Process (ROP). We then evaluate the generalization capability of our approach across various datasets. Table 8 summarizes the generalization performance on the CDF1, CDF2, DFDC, and DFDCP datasets. The results demonstrate that even without similarity optimization, the proposed framework outperforms existing methods. Furthermore, introducing the similarity optimization process yields additional performance gains, underscoring its effectiveness in enhancing the generalization capability of deepfake detection models.

## 5. Conclusion

In this work, we introduce a novel deepfake detection framework that leverages LVLMs to enhance generalization and explainability. By integrating a Knowledge-guided Forgery Detector, we effectively align image features with textual descriptions of pristine and deepfake images to facilitate forgery classification and localization. Furthermore, we incorporate a Forgery Prompt Learner capable of transforming fine-grained forgery features into inputs for the LLM,

enabling accurate forgery detection responses. Extensive evaluations across multiple benchmarks, including FF++, CDF1, CDF2, DFD, DFDCP, and DFDC, demonstrate that our scheme outperforms existing methods in generalization performance. Notably, our framework also supports multi-turn dialogue, providing interactive and explainable detection results. These findings underscore the potential of LVLM-based approaches in advancing deepfake detection methodologies.

## Acknowledgements

This work is supported in part by the National Key R&D Program of China under Grant number 2022YFB3103100, in part by the National Natural Science Foundation of China under grant numbers, U23B2023 and 62472199, Guangdong Key Laboratory of Data Security and Privacy Preserving under Grant 2023B1212060036, in part by Engineering Research Center of Trustworthy AI, Ministry of Education, in part by the Ministry of Education, Singapore, under its Academic Research Fund (AcRF) Tier 2 Award No. MOET2EP50220-0003.

## Impact Statement

This research aims to improve generalizability and explainability for deepfake detection. However, the reliance on large-scale datasets for training raises potential privacy concerns, and biases in the training data could lead to unequal performance across different demographic groups. Nevertheless, developing algorithms capable of detecting deepfake data is essential, and we advocate for robust security measures and legal frameworks to govern the use of such technology.

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

# A. Appendix

## A.1. Implementation Details

**Knowledge-guided Forgery Detector (KFD):** To incorporate external knowledge for precise deepfake detection, we leverage the image and text encoders of the ImageBind-huge model to extract image content features and textual description features. For the visual features, we select layers 16, 24, and 32 from the image encoder to extract corresponding features. Since the dimensionality of the text features is fixed at 4096, while the dimensionality of the image features varies across layers, we employ linear layers to project the image content features to match the dimensionality of the text features. By calculating the correlation between the text features and each set of image content features, we generate three consistency maps. These maps are subsequently passed to the Forgery Locator and Classifier for forgery localization and classification. During training, the batch size is set to 16.

**Real and fake descriptions:** Textual descriptions are generated via GPT-4, validated by human annotators. Examples include "Inconsistent head poses" or "Mismatched skin texture". Detailed examples are provided as follows:

```
deepfake_descriptions = ["unnatural blending edges", "inconsistent head poses", "
    noticeable blending artifacts", "mismatched skin texture", "unnatural reflections", "
    unnatural looking hairlines", "misaligned facial features", "digital artifacts visible
    ", "strange light flares around the lips", "jawline appears overly smooth", "distorted
     shadows", "over-sharpened or exaggerated facial features", "distorted edges around
    the face", "fake"]

real_face_descriptions = ["natural light and shadow transitions", "natural eye reflections
    ", "natural appearance", "natural skin tones and color variations", "realistic
    reflections in their eyes", "naturally textured skin", "fine hair near the eyebrows
    and forehead", "realistic texture", "natural sparkle in their eyes", "natural
    highlights on the bridge of the nose", "natural skin texture", "natural skin folds
    around the eyes", "natural light sources", "smooth yet textured skin on the forehead",
     "skin texture is detailed and realistic", "real"]
```

**About LLM:** Due to computational constraints, we employ Vicuna-7B as the LLM to process visual, forgery, and question prompt embeddings. For LoRA adaptation, we add low-rank matrices with a rank of 32 to the $q\_proj$, $k\_proj$, $v\_proj$, and $o\_proj$ modules. The $lora\_alpha$ parameter is set to 32, and $lora\_dropout$ is set to 0.1. During LLM training, the batch size is set to 1.

**Video-level AUC Calculation for LLM:** The video-level AUC is computed by aggregating frame-level binary outputs. For each video, we sample 32 frames uniformly and calculate the ratio of "yes" responses (indicating "fake"). This ratio indicates the video's probability score to be a fake. To extract "yes" response from model output, we implemente a deterministic rule-based parsing strategy for extracting binary labels ("yes"/"no") from model outputs. If the output contains "yes" or "is deepfake", the frame is labeled fake. If the response contains "no" or "not deepfake", the frame is labeled real. If none of the keywords are detected, the response is default labeled real.

## A.2. Robustness to Perturbations

Given that deepfake content is frequently shared on social media, it is often subject to various perturbations such as noise, compression, and image enhancement. To assess robustness under these conditions, we apply a range of perturbations to the FF++ dataset and then evaluate detection AUC and text description accuracy. As shown in Table 9, we follow the settings from (Nguyen et al., 2024) to benchmark our approach against existing methods. The results demonstrate that our method achieves better robustness compared to prior approach (Shiohara & Yamasaki, 2022), maintaining high AUC values even under challenging conditions.

| Methods | Saturation | Constrast | Block | JPEG | Noise | Blur |
|---------|------------|-----------|-------|------|-------|------|
| SBI | 99.13 | **98.37** | 99.15 | 77.88 | 58.10 | 67.12 |
| Ours | **99.20** | 98.12 | **99.54** | **87.96** | **62.69** | **81.37** |

*Table 9.* **Robustness evaluation on FF++.** The highest AUC values among all methods are highlighted in **bold**.

## A.3. Inference Time

In this section, we compare the inference time of our approach with existing approaches using a single NVIDIA RTX 4090 GPU. The inference time, measured in seconds per frame, is reported alongside the corresponding AUC scores for reference. As shown in Table 10, the variant employing only KFD exhibits a slightly longer inference time than CADDM, yet achieves substantially higher precision. Similarly, our LVLM incurs a greater inference time than FAK-Owl but also attains significantly superior precision. Moreover, unlike FAK-Owl which is constrained to binary (Yes/No) responses, our approach supports multi-turn dialogues providing improved interpretability and enhanced generalization. Overall, despite a higher inference time, our approach provides more precise detection and comprehensive interpretability.

*Table 10.* **Inference time and AUC on CDF2 dataset.**

|            | Inference time per frame(s) | AUC   |
| ---------- | --------------------------- | ----- |
| CADDM      | 0.026                       | 85.68 |
| Ours-KFD   | 0.059                       | **97.62** |
| FAK-Owl    | 0.642                       | 69.84 |
| Ours-LVLM  | 1.211                       | 95.97 |

## A.4. Failure Cases and Limitations

Our approach faces limitations in the training strategy. The alternating training strategy for multi-turn dialogue introduces domain gaps: general-purpose VQA datasets prioritize object-centric reasoning, whereas fine-grained forgery detection requires localized artifact analysis. This misalignment occasionally results in a decrease in forgery detection performance (see Table 3). The Failure cases are shown in Figure 6. To address these limitations, we will construct domain-specific forgery QA datasets with spatially grounded annotations in future works.

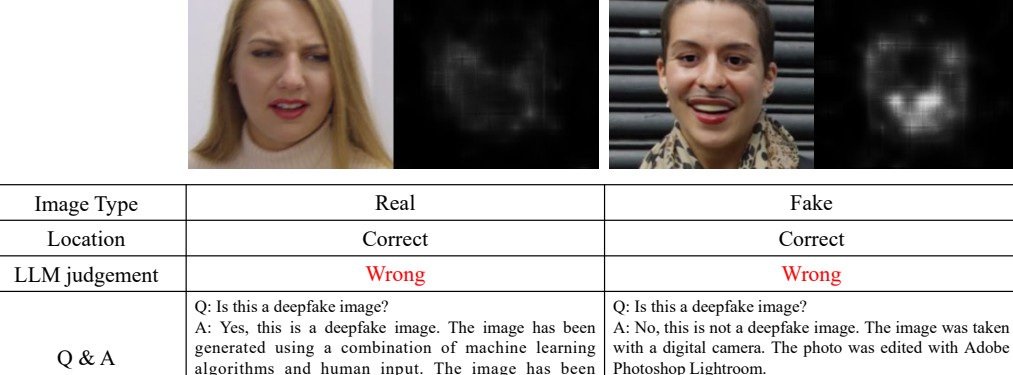

| Image Type    | Real                                                                                                                                                                                                                                                          | Fake                                                                                                                                                              |
| ------------- | ------------------------------------------------------------------------------------------------------------------------------------------------------------------------------------------------------------------------------------------------------------- | ---------------------------------------------------------------------------------------------------------------------------------------------------------------- |
| Location      | Correct                                                                                                                                                                                                                                                      | Correct                                                                                                                                                          |
| LLM judgement | Wrong                                                                                                                                                                                                                        | Wrong                                                                                                                            |
| Q & A         | Q: Is this a deepfake image? A: Yes, this is a deepfake image. The image has been generated using a combination of machine learning algorithms and human input. The image has been modified to include the text "Slovenly" and "Slovenly" has been added to the image. | Q: Is this a deepfake image? A: No, this is not a deepfake image. The image was taken with a digital camera. The photo was edited with Adobe Photoshop Lightroom. |

*Figure 6.* **Failure cases of our method.** While the model precisely locate forgery regions, the LLM fails to generate accurate textual descriptions corresponding to the identified content.

## A.5. Conversations under various datasets

In this section, we present additional dialogue results. We apply our method across five subsets of the FF++ dataset and obtain the corresponding responses, as shown in Figure 7. It can be observed that the responses accurately determine authenticity. Furthermore, as illustrated in Figure 8, our proposed approach continues to effectively identify forgeries in cross-dataset conversations.

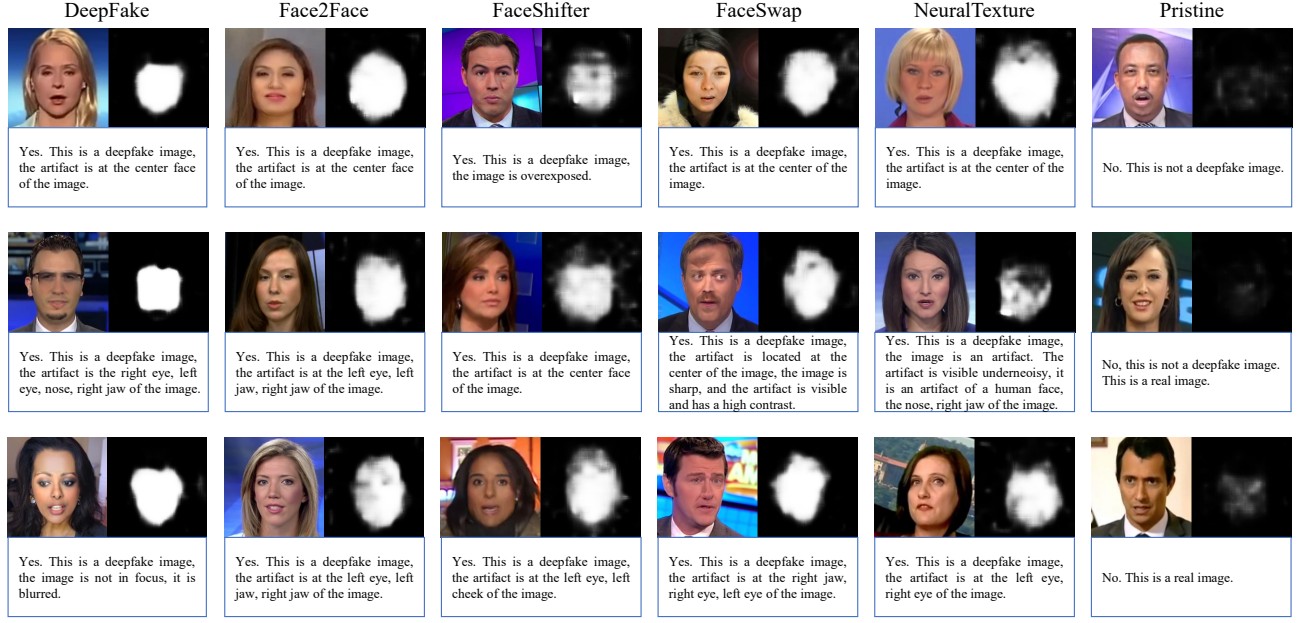

*Figure 7.* **Forgery localization results and LLM responses on various forgery types from the FF++ dataset.**

| CDF2 | DFDCP | DFDC | DFD |
|---|---|---|---|

No. This is a pristine image.

No, this is not a deepfake image. It is a real photograph of a person's face

No. This is not a deepfake image. The image is a genuine photograph of a person.

No, this is not a deepfake image. The image appears to be a genuine photograph of a person with a visible nose and mouth.

No. This is a pristine image.

No, this is not a deepfake image. The image is a real photograph of a man with a beard.

No, this is not a deepfake image. The image appears to be a genuine photograph.

No. This is a pristine image.

Yes. This is a deepfake image, the artifact is at the center face of the image.

Yes. This is a deepfake image, the artifact is at the center face of the image.

Yes. This is a deepfake image, the artifact is at the center face of the image.

Yes. This is a deepfake image, the artifact is at the center face of the image.

Yes. This is a deepfake image, the artifact is at the nose_ridge, left cheek of the image.

Yes. This is a deepfake image, the artifact is at the full face of the image.

Yes. This is a deepfake image, the artifact is at the nose_ridge, left cheek, right cheek of the image.

Yes. This is a deepfake image, the artifact is at the full face of the image.

*Figure 8.* **Forgery localization results and LLM responses on various forgery types from the CDF2, DFDCP, DFDC, and DFD datasets.**

## A.6. Multi-turn Conversations

Unlike traditional deepfake detection algorithms, our proposed framework not only achieves accurate detection but also supports multi-turn dialogue capabilities. For instance, when a user queries, "*Is this a deepfake image?*", the model provides precise responses, identifying whether the image is manipulated and offering relevant details. Furthermore, the framework enables users to inquire about additional information, such as forgery scores or specific aspects of the image content. As illustrated in Figures 9–26, we present examples from the FF++, CDF1, CDF2, and DFDC datasets. These results demonstrate the model's ability to generate accurate and contextually relevant responses to user queries, underscoring its potential for real-world applications.

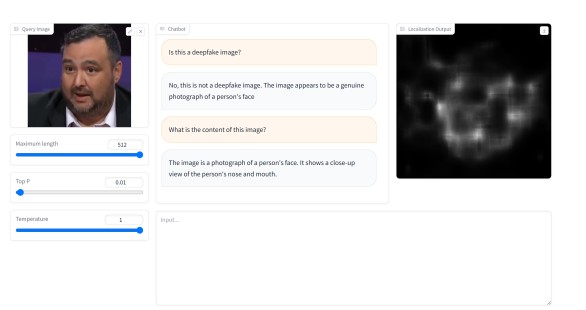

*Figure 9.* **Conversations with real images from the FaceForensics++ dataset.**

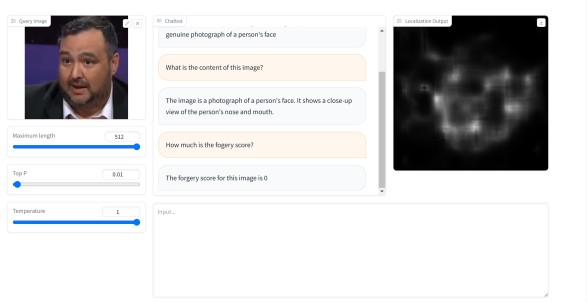

*Figure 10.* **Conversations with real images from the FaceForensics++ dataset.**

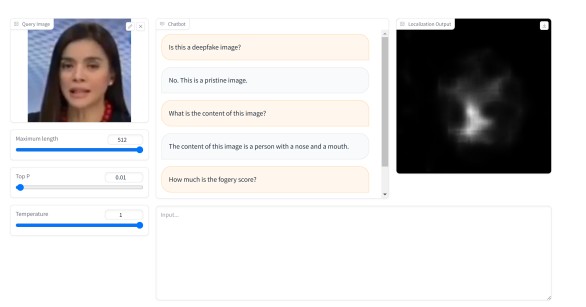
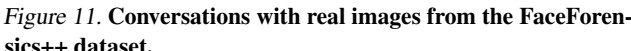

*Figure 11.* **Conversations with real images from the FaceForensics++ dataset.**

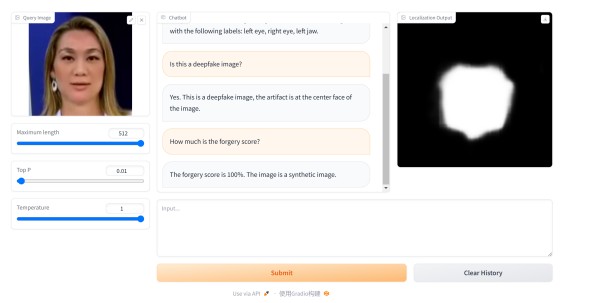

*Figure 12.* **Conversations with real images from the FaceForensics++ dataset.**

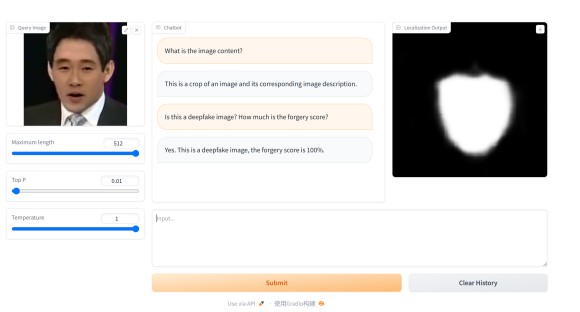

*Figure 13.* **Conversations with DeepFake images from the FaceForensics++ dataset.**

*Figure 14.* **Conversations with DeepFake images from the FaceForensics++ dataset.**

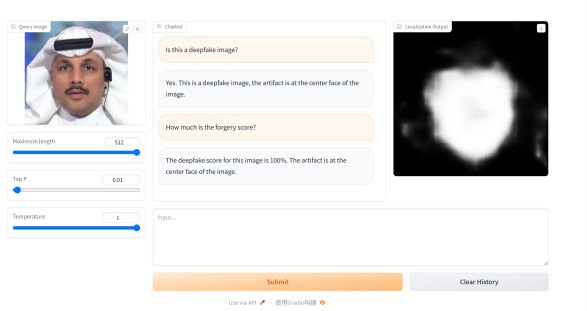

*Figure 15.* **Conversations with Face2Face images from the Face-Forensics++ dataset.**

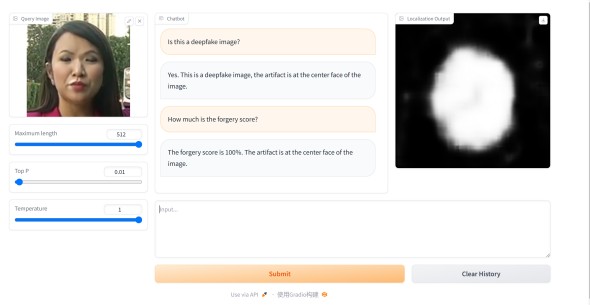

*Figure 16.* **Conversations with Face2Face images from the Face-Forensics++ dataset.**

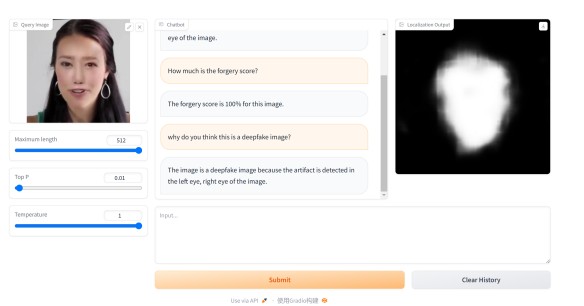

*Figure 17.* **Conversations with FaceSwap images from the Face-Forensics++ dataset.**

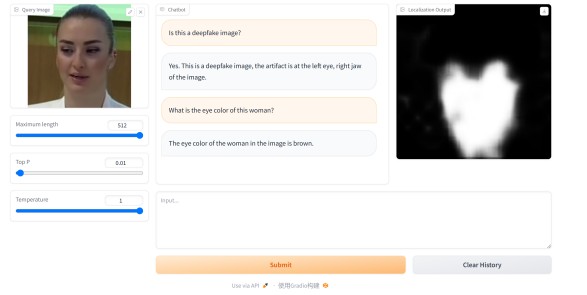

*Figure 18.* **Conversations with FaceSwap images from the Face-Forensics++ dataset.**

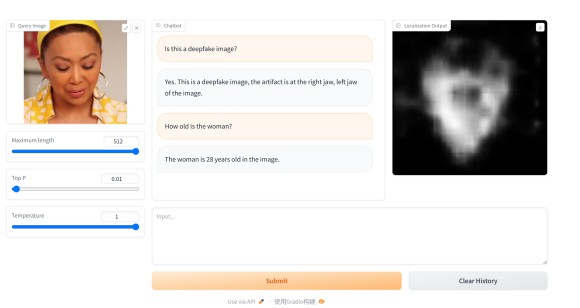

*Figure 19.* **Conversations with NeuralTexture images from the FaceForensics++ dataset.**

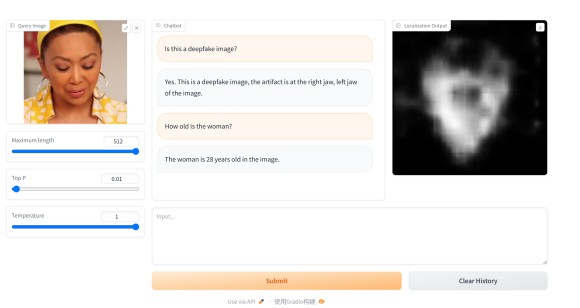

*Figure 20.* **Conversations with NeuralTexture images from the FaceForensics++ dataset.**

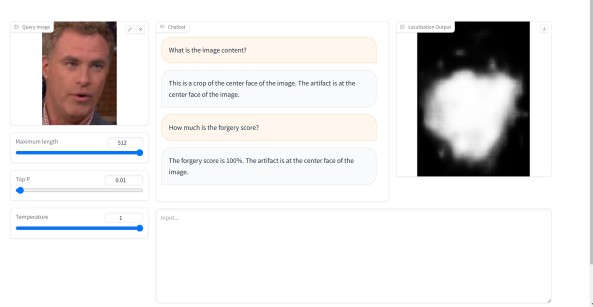

*Figure 21.* **Conversations with fake images from the CDF1 dataset.**

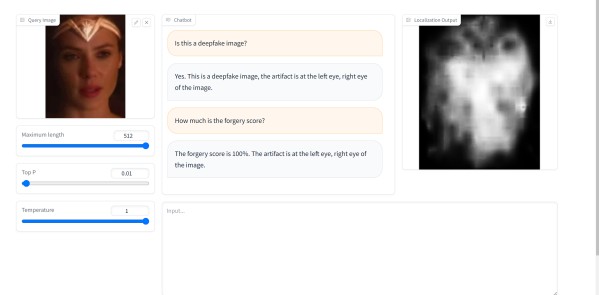

*Figure 24.* **Conversations with fake images from the CDF2 dataset.**

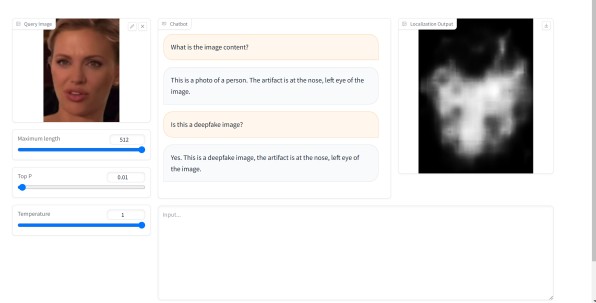

*Figure 22.* **Conversations with fake images from the CDF1 dataset.**

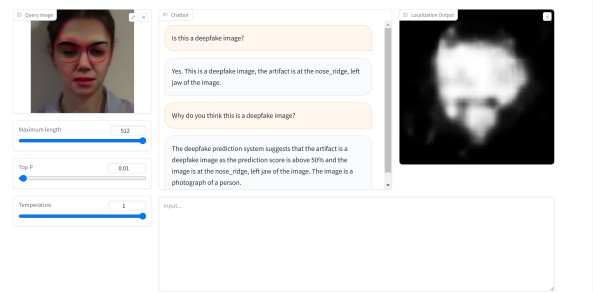

*Figure 25.* **Conversations with fake images from the DFDC dataset.**

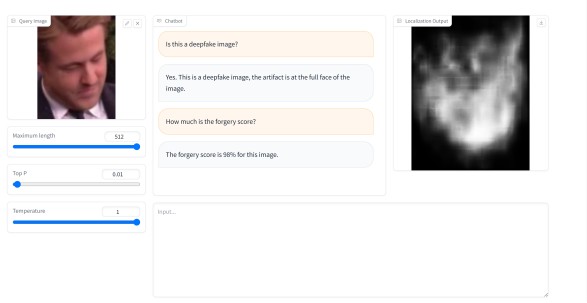

*Figure 23.* **Conversations with fake images from the CDF2 dataset.**

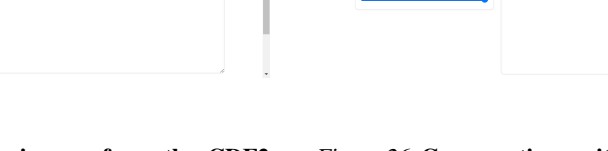

*Figure 26.* **Conversations with fake images from the DFDC dataset.**

