# OpenReview forum: "Unlocking the Capabilities of Large Vision-Language Models for Generalizable and Explainable Deepfake Detection"
_ICML.cc/2025/Conference — ICML 2025 poster_

### Official Review · Reviewer_Led7 · 2025-03-08

**Overall Recommendation:** 2

**Summary:**

This paper proposes a novel framework that leverages Vision-Language Models (VLMs) for deepfake detection, addressing their current limitations in forensic analysis. The core innovation is a three-component approach: (1) a knowledge-guided forgery adaptation module that aligns VLM semantic space with forensic features through contrastive learning, (2) a multi-modal prompt tuning framework that optimizes visual-textual embeddings for improved localization and explainability, and (3) an iterative refinement strategy enabling evidence-based reasoning through multi-turn dialog. The architecture integrates a Knowledge-guided Forgery Detector with a Large Language Model, allowing the system to not only detect and localize deepfakes across diverse manipulation types but also provide natural language explanations about the detected forgeries, significantly advancing both generalizability and interpretability in deepfake detection.

**Claims And Evidence:**

The paper's claims are partially supported by evidence, with several limitations:

1. The multi-turn dialogue capability claim is prominently featured in the abstract and conclusion but receives minimal validation in the main paper. Figure 5 shows only single-turn examples, and the paper notes that "Additional multi-turn dialogue examples are provided in the supplementary material." However, no appendix or supplementary material is found. Without these examples, this key claimed contribution lacks sufficient evidence.

2. The "knowledge-guided forgery adaptation module" lacks clear documentation of what specific external manipulation knowledge is incorporated. While Section 3.1 describes the mechanism, it doesn't detail the nature of the textual descriptions ($D _ {real}$ and $D _ {fake}$) that serve as the knowledge source.

3. The forgery localization capability is demonstrated qualitatively through GradCAM visualizations in Figure 4, but no quantitative evaluation of localization accuracy is provided, making it difficult to objectively assess this capability.

4. The ablation study in Table 8 evaluates the Reference-based Optimization Process but doesn't fully isolate the contribution of each of the three claimed novel components, particularly the multi-modal prompt tuning framework's specific impact.

5. The data simulation process described in Section 3.3 is used for training, but there's insufficient evaluation of how well this simulated data represents real-world deepfakes, which could affect generalization claims.

**Essential References Not Discussed:**

The paper adequately cites relevant related works, with no major omissions.

**Experimental Designs Or Analyses:**

I examined several aspects of the experimental methodology, finding both strengths and limitations:

1. The dataset selection and evaluation protocols using standard benchmarks (FF++, CDF2, DFD, DFDCP, DFDC) with established metrics (AUC, AP) follow sound practices in the field.

2. The cross-dataset evaluation appropriately tests generalization capability, training on FF++ real data and testing across multiple datasets.

3. However, the LLM evaluation process lacks methodological clarity. The paper states they "utilize the LLM's output ('Yes' or 'No') to classify authenticity" but doesn't explain how they handle cases where LLM outputs may be nuanced or ambiguous rather than strictly binary.

4. The LVLM-based method comparison in Table 3 shows baseline performances that are surprisingly low (e.g., Qwen2-VL at ~48% AUC on some datasets, near random chance), raising questions about implementation fairness or model configuration.

5. The training data simulation using Poisson blending of affine-transformed images (Section 3.3) may not adequately represent artifacts found in modern AI-generated deepfakes, yet this potential limitation isn't acknowledged.

6. The ablation studies, while informative, don't systematically isolate the contribution of each of the three claimed core components, particularly for the multi-modal prompt tuning framework.

**Methods And Evaluation Criteria:**

The proposed methods generally align with the deepfake detection problem, but several evaluation aspects raise concerns:

1. For evaluation datasets, the authors appropriately use standard benchmarks (FF++, CDF2, DFD, DFDCP, DFDC) and conventional metrics (AUC and AP), which is reasonable for comparative assessment.

2. The cross-dataset and cross-manipulation evaluations are particularly appropriate for testing generalization capabilities, which is a critical challenge in deepfake detection.

3. However, for a framework claiming explainability as a key contribution, there's no quantitative evaluation of explanation quality. While textual outputs are shown in Figure 5, no metrics assess whether these explanations accurately identify the manipulation technique or forgery characteristics.

4. For the claimed localization capability, only qualitative GradCAM visualizations are provided without quantitative localization accuracy metrics, making it difficult to objectively compare with other localization approaches.

5. The paper emphasizes multi-turn dialogue as a key capability. However, it neither establishes a standardized evaluation protocol to measure its effectiveness in deepfake analysis nor provides any examples of such interactions.

6. The forgery data simulation approach for training relies on Poisson blending of affine-transformed real images, but there's limited analysis of whether this adequately represents the artifacts found in modern AI-generated deepfakes, potentially limiting real-world applicability.

**Other Comments Or Suggestions:**

1. **Typos and clarifications needed:**
   - Abstract: "due to the misaligned of their knowledge" should be "misalignment"
   - Section 3.2: "$E _ {forgery} \in \mathbb{R}^{{n _ f} \times C _ {emb}}$" - $n _ f$ is not properly defined

2. **Technical clarifications needed:**
   - The "learnable context" mentioned in Section 3.1 needs more detailed explanation
   - The specific format of prompts used during LLM evaluation should be explicitly shown
   - The process for converting LLM outputs to binary decisions for AUC calculation requires clarification

3. **Evaluation suggestions:**
   - Include quantitative metrics for localization accuracy
   - Provide more examples of multi-turn dialogues in the main paper
   - Compare computational efficiency and inference time with existing methods
   - Consider human evaluation for explanation quality

4. **Missing discussion:**
   - Analysis of potential failure cases would strengthen the paper

**Other Strengths And Weaknesses:**

**Strengths:**
1. The integration of VLMs with deepfake detection addresses a genuine need for improved generalization and explainability in forensic analysis.
2. The knowledge-guided approach acknowledges an important gap in current methods: the difficulty of capturing human forensic knowledge through data augmentation alone.
3. The reference-based optimization process shows promising results for enhancing feature robustness.
4. The paper demonstrates versatility by evaluating both cross-dataset and cross-manipulation scenarios, which is crucial for real-world applications.

**Weaknesses:**
1. The performance improvements, while consistent, are relatively modest (average 1.34% AUC improvement) given the complexity of the proposed framework.
2. The technical description of the knowledge acquisition process is vague --- specifically how the "learnable context" is integrated with predefined real/fake descriptions.
3. The framework's complexity (multiple interconnected components) may hinder practical deployment compared to simpler approaches.
4. The paper doesn't adequately analyze failure cases or where the approach struggles, which would provide valuable insight into its limitations.
5. The training data simulation approach may not capture the sophisticated artifacts produced by state-of-the-art deepfake generators, potentially limiting real-world effectiveness.
6. The lack of quantitative metrics for both localization accuracy and explanation quality makes it difficult to objectively assess two of the framework's key claimed capabilities.

**Questions For Authors:**

1. Could you provide quantitative evaluation metrics for the localization accuracy of your approach? The current paper only shows qualitative visualization without objective metrics to compare against other localization-capable methods.

2. What specific "external manipulation knowledge" is incorporated into your framework? Section 3.1 mentions "real and fake image descriptions" but doesn't detail their content or source, making it difficult to assess this key component.

3. The multi-turn dialogue capability is prominently claimed but not demonstrated. Could you explain how you evaluate dialogue quality, and how your approach specifically enables multi-turn reasoning beyond what existing LVLM methods provide?

4. In Table 3, several baseline LVLM methods (e.g., Qwen2-VL at 47.99% average AUC) perform near random chance. What explanation do you have for these unexpectedly low baselines?

5. How does your forgery data simulation approach using Poisson blending of affine-transformed images capture the artifacts produced by modern AI-generated deepfakes? This seems critical for real-world generalization.

**Relation To Broader Scientific Literature:**

This paper's contributions relate to several research streams:

1. **Deepfake detection methods**: Traditional approaches (Li et al., 2020; Shiohara & Yamasaki, 2022; Nguyen et al., 2024) focused on data augmentation, feature consistency, and frequency domain analysis. This work acknowledges their limitations in capturing human knowledge about forgery characteristics and proposes VLMs as a solution.

2. **Vision-Language Models**: Builds upon recent LVLM architectures like BLIP-2 (Li et al., 2023), LLaVA (Liu et al., 2024), and MiniGPT-4 (Zhu et al., 2024), but adapts them specifically for forensic analysis --- a departure from their general image understanding focus.

3. **Multimodal forensics**: Extends work like FakeShield (Xu et al., 2024) and FKA-Owl (Liu et al., 2024b), which also applied LVLMs to forgery detection, by introducing the knowledge-guided forgery detector and incorporating detailed localization capabilities.

4. **Prompt tuning literature**: The forgery prompt learning approach builds upon prompt tuning techniques (Lester et al., 2021; Liu et al., 2022) but adapts them for the multimodal forensic context.

5. **Explainable AI**: While works like FFAA (Huang et al., 2024) also explored explainable forgery analysis, this paper's integration of multi-turn dialogue capabilities represents an evolution in interactive forensic analysis.

**Theoretical Claims:**

This paper does not present significant theoretical claims requiring formal mathematical proofs. The work is primarily empirical, focusing on the design and evaluation of a VLM-based framework for deepfake detection.

The mathematical formulations presented in the paper (Equations 1-6) employ standard techniques commonly used in machine learning:
- Equations 1-2 describe similarity computations between visual and textual features
- Equation 3 uses the established Dice loss for segmentation
- Equation 4 employs standard binary cross-entropy loss for classification
- Equation 5 uses cross-entropy loss for the LLM
- Equation 6 describes a basic blending process for generating training data

These formulations appear correctly applied for their intended purposes, but they don't constitute novel theoretical contributions requiring verification of mathematical proofs.

---

> ### Author Rebuttal · Authors · 2025-04-01
>
> **1) Multi-turn Dialogue Capabilities:** See response to KySN Q1.
>
> **2) Learnable Context:** Textual descriptions are generated via GPT-4, validated by human annotators. Examples include “Inconsistent head poses” or “Mismatched skin texture”. These annotations are available in https://anonymous.4open.science/r/DFDGPT-8E5C.
>
> **3) Quantitative Evaluation of Localization Accuracy and Explanation Quality:** We thank the reviewer for the valuable suggestion. As suggested, we have conducted a quantitative evaluation for both the localization and explanation capabilities of our approach using two metrics: Text Localization Accuracy (TLA) and Cosine Semantic Similarity (CSS). TLA measures the consistency between the tampered region descriptions produced by our LLM and the ground-truth localization annotations, by using the Dice coefficient. To objectively assess the explanation quality, following FakeShield (ICLR’25), we calculate the CSS by computing the cosine similarity between the high-dimensional semantic vector representations of the generated explanation text and the corresponding ground-truth text.
> For this evaluation, we trained both our approach and a fine-tuned version of PandaGPT on our synthetic forgery dataset. The results, summarized in the table below, indicate that our method achieves significantly higher localization accuracy and explanation quality than PandaGPT. We appreciate the reviewer's suggestion, and we will incorporate these quantitative evaluation metrics and the corresponding results into the revised manuscript.
>
> || CDF1     |         | CDF2    |         | DFDC    |         | DFDCP   |         |
> |:--------:|:--------:|:-------:|:-------:|:-------:|:-------:|:-------:|:-------:|:-------:|
> |          | TLA     | CSS     | TLA     | CSS     | TLA     | CSS     | TLA     | CSS     |
> | PandaGPT | 0.6239  | 0.7666  | 0.6241  | 0.7717  | 0.6389  | 0.7606  |  0.6220 | 0.7846  |
> | Ours     | 0.7762  | 0.8532  | 0.7755  | 0.8498  | 0.7662  | 0.8235  | 0.7842  | 0.8370  |
>
> **4) Component Contribution:** We appreciate the opportunity to clarify the contributions of each module in our framework. In Table 5, we present ablation experiments on our three primary modules, which validate their individual effectiveness. The Reference-based Optimization Process (ROP) is specifically designed to enhance the training stability and feature discriminability of the Knowledge-guided Forgery Detector (KFD). To isolate its contribution, we compare its performance with and without ROP. The ROP is not directly connected to other components (LLM and LoRA), and its benefits are implicitly propagated through KFD’s refined features during training. We will explicitly clarify ROP’s role in Section 3.1 to avoid ambiguity.
>
> **5) Simulation Limitations:** See response to MUQX Q4.
>
> **6) AUC Calculation by LLM Output:** See response to MUQX Q1.
>
> **7) Implementation Fairness about Qwen:** We note that Qwen-VL is a general-purpose visual question answering model and is not specifically designed for deepfake detection, which leads to its lower performance. To ensure fairness, all methods are evaluated under identical pre-processing conditions (32 frames per video, 224×224 resolution) and consistent evaluation protocols. Furthermore, we will release our code to facilitate reproducibility.
>
> **8) Inference Time:** See response to KySN Q5.
>
> **9) Failure Cases and Limitations:**  Our approach faces limitations in the training strategy. The alternating training strategy for multi-turn dialogue introduces domain gaps: general-purpose VQA datasets prioritize object-centric reasoning, whereas fine-grained forgery detection requires localized artifact analysis. This misalignment occasionally results in a decrease in forgery detection performance (see Table 2). The Failure cases are available in https://anonymous.4open.science/r/DFDGPT-8E5C. To address these limitations, we will construct domain-specific forgery QA datasets with spatially grounded annotations. We will add a discussion section to elaborate limitations and future works.
>
> **10) SOTA Generators Tested:** We evaluated various deepfakes generated by state-of-the-art (SOTA) models, including StyleGAN-3, DiT-XL/2, Stable Diffusion, etc. For experimental results, refer to the response to MUQX Q3.
>
> We thank all reviewers for their constructive feedback. Revisions will address every point rigorously.

---

> > ### Comment · Reviewer_Led7 · 2025-04-07
> >
> > I appreciate the authors’ detailed response during the rebuttal phase. Although it improved my understanding in some areas, it does not sufficiently shift the overall strength or novelty of the submission to warrant a change in score.

---

> > > ### Author Response · Authors · 2025-04-08
> > >
> > > We appreciate the reviewer’s acknowledgment of our detailed responses during the rebuttal phase. We respectfully request a further consideration of our contributions, which we believe offer significant novelty and strength in several key areas:
> > >
> > > **Comprehensive Multi-Modal Framework:** Our work introduces a novel LVLM-based deepfake detection framework that integrates three key components: a knowledge-guided forgery detector, a multi-modal prompt tuning mechanism, and an iterative refinement strategy for multi-turn dialogue. Unlike previous methods that focus solely on spatial or frequency-domain artifacts, our framework leverages external forensic knowledge and fine-grained prompt embeddings to bridge the gap between visual cues and textual descriptions. This architecture enables our model not only to classify images as real or fake but also to generate localized, human-readable explanations of forgery regions.
> > >
> > > **Visual-Textual Consistency for Deepfake Detection:** Our approach capitalizes on the strong visual–textual representations learned by large-scale pretrained models, specifically the CLIP visual encoder within the ImageBind. CLIP, trained on billions of image–text pairs, inherently captures rich semantic and fine-grained visual features. We leverage this capability by aligning the visual features extracted from input images with corresponding textual embeddings that describe pristine and potentially manipulated content. After fine-tuning with SBI image–text pairs, our approach is able to detect deepfake artifacts accurately.
> > >
> > > **Robustness Across Diverse Forgery Scenarios:** We have conducted extensive experiments on multiple benchmarks, including FF++, CDF1, CDF2, DFD, DFDCP, DFDC, and DF40. Our approach achieves state-of-the-art AUC values under cross-dataset evaluations. Moreover, our approach effectively handles various forgery methods, ranging from conventional face-swapping to entirely synthesized forgeries. In doing so, it substantially advances the current performance envelope of deepfake detection techniques.
> > >
> > > **Explainability and Multi-Turn Dialogue Capability:** Beyond detection accuracy, our approach supports multi-turn dialogues that allow users to inquire about specific image content and forgery regions. This interactive capability not only enhances transparency but also contributes to the overall explainability of the detection process—a critical need in forensic applications. Our extensive qualitative and quantitative evaluations (e.g., through GradCAM visualizations, CSS metric, and video-level AUC/AP metrics) further substantiate this contribution.
> > >
> > > In summary, we believe that the integration of multi-modal forensic knowledge, the visual-textual consistency, and the novel multi-turn dialogue capability collectively represent a significant step forward in the field of deepfake detection. We respectfully hope that these clarifications will prompt a favorable re-assessment of our submission.
> > >
> > > Thank you for your consideration.

---

### Official Review · Reviewer_MUQX · 2025-03-11

**Overall Recommendation:** 3

**Summary:**

This paper proposes leveraging LLM and VLM to improve the model generalization and explainability. It is achieved by a two-stage pipeline: A Knowledge-guided Detection using humans prior to generating feature embedding; leveraging these embedding for LLM to output detection results. The experimental results show that it successfully incorporates the capacity of LLM into deepfake detection.

**Claims And Evidence:**

Yes

**Essential References Not Discussed:**

Please refer to Experimental Designs Or Analyses.

**Experimental Designs Or Analyses:**

It is partially reasonable. I have a few suggestions.
1. Fig. 5 is a crucial experiment to validate one of the most important advantages of using LLM&VLM, that is, its explainability. Hence, it should be conducted on a larger scale, at least not limited to the in-dataset scenarios. In addition, you should have appropriate strategies for evaluating the accuracy of the generated text, otherwise, how do you know if the output text is appropriate?
2. More recent SoTA should be compared. For example, [1] [2] [3]. Among them, [1] also discusses the usage of SBI, which is deployed in this paper.
3. More datasets are recommended. The applied datasets in Tab. 1 are indeed only three types, i.e., CDF, DFDC, and DFD. You may include more datasets like DF40 for better illustrations.





[1] Can We Leave Deepfake Data Behind in Training Deepfake Detector?// NIPS24

[2] Exploring Unbiased Deepfake Detection via Token-Level Shuffling and Mixing // AAAI25

[3] DiffusionFake: Enhancing generalization in deepfake detection via guided stable diffusion //NIPS24

**Methods And Evaluation Criteria:**

Yes, it makes sense.

**Other Comments Or Suggestions:**

**update after rebuttal**
The authors have partially addressed my concern, therefore I retain my original rating.

**Other Strengths And Weaknesses:**

The method employed is rather simple, essentially following a contrastive learning approach. However, the problem it addresses is intriguing and holds practical significance. My primary concern is that the experiments may be somewhat insufficiently conducted， please refer to Experimental Designs Or Analyses.

**Questions For Authors:**

Notably, SBI can only simulate the face-blending artifacts. Therefore, it cannot provide forgery clues about generative artifacts. How can your method deal with the fake image generated by Entire-face Synthesis without blending?

**Relation To Broader Scientific Literature:**

It is relevant to the generalizable deepfake detection.

**Theoretical Claims:**

N/A

---

> ### Author Rebuttal · Authors · 2025-04-01
>
> **1)AUC Calculation by LLM Output:** To ensure rigorous and reproducible evaluation of text-level AUC, we implemented a deterministic rule-based parsing strategy for extracting binary labels ("yes"/"no") from model output. If the output contains "yes" or "is deepfake", the frame is labeled fake. If the response contains "no" or "not deepfake", the frame is labeled real. If none of the keywords are detected, the response is default labeled real. The experimental results in Table 3,4,5,7 prove the effectiveness of our scheme.
>
> **2)Cross-dataset evaluation of LLM&VLM:** We appreciate the suggestion. A figure illustrating the cross-dataset evaluation has been added to the anonymous GitHub repository and is available at https://anonymous.4open.science/status/DFDGPT-8E5C. We will incorporate this figure into Section 4.3.
>
> **3) Comparison with Recent SOTA:**  Similar to Table 2, we add more comparisons against recent SOTA detection methods across various test sets.  The new results are listed in Table A below and will be incorporated into our manuscript. ProDet is evaluated using publicly available code, while the results for other approaches are obtained from their original publications.
>
> Table A. Generalization Performance across various datasets.
>
> |Methods|Venue|CDF2||DFDC||CDF1||DFDCP||
> |-|-|-|-|-|-|-|-|-|-|
> |||AUC|AP|AUC|AP|AUC|AP|AUC|AP|
> |ProDet|NIPS’24|92.62|96.05|71.52|72.8|94.48|96.66|82.83|88.89|
> |RepDFD|AAAI’25|89.94|-|80.99|-|-|-|95.03|-|
> |CFM|TIFS’24|89.65|-|80.22|-|-|-|-|-|
> |ED|AAAI’24|93.6|-|75.4|-|-|-|90.2|-|
> |UDD|AAAI’25|93.13|-|81.21|-|-|-|88.11|-|
> |Ours|-|94.71|93.59|79.12|77.69|97.62|97.67|91.81|88.26|
>
> **4) More Datasets:** As suggested, we have used the DF40 dataset to further evaluate the generalization capability of our approach. The DF40 dataset comprises several synthetic deepfake datasets that are generated using real images in FF++. We use these datasets to evaluate our method’s cross-manipulation detection ability. Notably, our approach continues to exhibit robust detection performance against state-of-the-art deepfake generation techniques (e.g., FSGAN, E4S, LIA, StyleGAN, DDIM, PixArt-α, etc.).
>
> Table B. Generalizaiton performance across various deepfake methods. The models are all trained on the Faceforensic++ dataset and then evaluated on unseen types of deepfakes.
> |Face-swapping||||||||
> |:-:|:-:|:-:|:-:|:-:|:-:|:-:|:-:|
> ||UniFace|SimSwap|InSwapper|FSGAN|FaceDancer|BlendFace|e4s|FaceSwap|
> |SBI|89.02|93.22|88.52|89.62|78.18|95.09|86.36|94.37|
> |CADDM|86.86|90.41|78.65|88.86|76.54|90.75|87.92|97.96|
> |Ours|90.61|90.97|87.64|93.75|82.97|92.10|94.68|93.34|
> |Face-reenactment|||||||||
> ||PIRender|OneShot|HyperReenact|FOMM|FS_vid2vid|TPSMM|MCNet|LIA|
> |SBI|81.81|87.54|65.31|88.05|83.72|82.13|83.47|89.22|
> |CADDM|77.37|85.05|69.26|84.77|72.86|71.35|73.40|69.67|
> |Ours|88.29|90.45|81.55|93.34|71.56|78.14|81.48|99.99|
> |EntireFaceSynthesis|||||||||
> ||VQGAN|StyleGAN2|StyleGAN3|StyleGAN-XL|DDIM|DiT-XL/2|PixArt-α|RDDM|
> |SBI|91.50|97.91|97.91|23.26|99.56|79.04|98.78|53.66|
> |CADDM|99.99|100.00|100.00|98.69|98.10|79.90|99.74|98.59|
> |Ours|99.99|100.00|100.00|100.00|99.93|94.18|100.00|86.59|
>
> **5) Limitations about Simulation:** Yes. The SBI-based synthetic forgery pipeline is primarily designed to simulate face-blending artifacts by focusing on modeling boundary inconsistencies. Although it is effective in detecting certain forgeries, its performance degrades when applied to fully synthesized images generated by models such as StyleGAN-XL and RDDM, as evidenced by the experimental results in our response to Q3. While our approach is built on SBI, the integration of pre-trained knowledge from LVLM has notably enhanced its generalized detection ability across various types of forgeries (see Table B). In future work, we will incorporate model-related artifacts into the forgery generation process to further improve detection performance. We will include a dedicated discussion section to further elaborate on these limitations and outline potential directions for future work.

---

> > ### Comment · Reviewer_MUQX · 2025-04-02
> >
> > Thanks for your rebuttal. I have a few questions:
> >
> > 1. AUC Calculation by LLM Output
> > According to your statement, the final prediction is obtained through a deterministic rule-based parsing strategy, resulting in binary output. In other words, your prediction lacks confidence scores, which may introduce a significant issue. First, is it not the case that the LLM might generate ambiguous judgments, such as “Maybe it is a deepfake, but I'm not sure”? Would this lead to a misalignment between the final results and the LLM output? Secondly, could the absence of confidence scores prevent the model from distinguishing between easy and hard samples, thus impairing its understanding capability? Finally, without confidence scores, with only 0 and 1 as outputs, it would be impossible to utilize different thresholds to plot the ROC curve in terms of FPR and TPR. Similarly, it would not be feasible to plot the PR curve. Consequently, how do you calculate the AUC and AP?
> >
> > 2. In Table A, the AUC for "Ours" is the best, but the AP is lower than that of ProDet. Why is this the case?
> >
> > 3. The learned DFD capability of your LLM model during training entirely depends on the blending clues generated by the SBI. In other words, the LLM has never seen any model-based synthetic artifacts. How, then, does it learn to differentiate model-based artifacts?

---

> > > ### Author Response · Authors · 2025-04-05
> > >
> > > Thanks for your comment. Below are our detailed responses.
> > >
> > > **(1) AUC Calculation by LLM Output:** Although our LLM prediction does not provide a confidence score for a single frame, we can still compute the forgery confidence score for each video by calculating the proportion of frames that are classified as fake. Below is our detailed response to the reviewer’s concerns:
> > >
> > > 1)	**Ambiguous Judgments:** Our experiments on 7,000 images from the CDFv2 dataset have shown that our system does not produce ambiguous judgments. During fine-tuning, we enforced a standardized response protocol during model fine-tuning, which guides the LLM to generate clear, binary outputs. Although the LLM is capable of producing ambiguous statements in principle, our controlled fine-tuning has significantly minimized such occurrences, and no ambiguous outputs were observed in our experiments.
> > >
> > > 2)	**Absence of Confidence Scores:** At the frame level, the LLM outputs a binary decision; however, the associated Vision-Language Model (VLM) is capable of generating a continuous confidence score for each frame. Moreover, we can incorporate this frame-level confidence information into the LLM through a text-based Q&A process (evidenced by our screenshots in the supplementary material). For video-level evaluation, we calculate the forgery probability as the ratio of frames classified as fake, yielding a continuous score in the range [0,1]. This video-level confidence score enables us to compute ROC and PR curves, thereby allowing us to accurately calculate both AUC and AP.
> > >
> > > 3)	**AUC and AP Calculation:** Our evaluation metrics (AUC and AP) are calculated at the video level. For each video, we uniformly sample 32 frames and define the video’s forgery probability as the proportion of frames classified as fake. Consequently, each video receives a confidence score in the range [0, 1], which allows us to compute AUC and AP accurately.
> > >
> > > **(2) Discrepancy between AUC and AP:** In Table A, our method achieves the highest AUC while exhibiting a slightly lower AP compared to ProDet. We attribute this discrepancy primarily to the differences in how these metrics are calculated. AUC (Area Under the ROC Curve) weighs all false positives equally, whereas AP (Average Precision) weighs false positives at a threshold $\tau$ with the inverse of the model’s likelihood of outputting any scores greater than $\tau$ [1]. This phenomenon, where a method shows high AUC but relatively lower AP, has also been observed in other studies [2, 3], further underscoring that the two metrics capture different aspects of detection performance.
> > >
> > > [1] McDermott, M., Zhang, H., Hansen, L., Angelotti, G., & Gallifant, J. (2024). A closer look at auroc and auprc under class imbalance. NIPS, 37, 44102-44163.
> > >
> > > [2] Nguyen, Dat, et al. "Laa-net: Localized artifact attention network for quality-agnostic and generalizable deepfake detection." CVPR. 2024.
> > >
> > > [3] Yan, Zhiyuan, et al. "Transcending forgery specificity with latent space augmentation for generalizable deepfake detection." CVPR. 2024.
> > >
> > > **(3) Model-Based Artifacts:** Our approach generalizes beyond the blending cues generated by the SBI process due to three key aspects:
> > >
> > > 1)	**Blending Operations:** The SBI pipeline inherently incorporates blending operations. This allows our model to effectively detect forgeries in datasets like FF++, CDF, and DFDC, where blending is a common post-processing operation.
> > >
> > > 2)	**Convolution and Up-sampling–like operations in SBI:** In generating the SBI dataset, we apply various image processing techniques such as blurring and scaling. These operations involve convolution and up-sampling, which can introduce artifacts similar to those found in model-based synthetic images. This exposure helps the model learn discriminative features that extend beyond simple blending clues and mimic the artifacts typically seen in fully synthesized images.
> > >
> > > 3)	**Pretrained CLIP Visual Encoder:** Our approach leverages the CLIP visual encoder from ImageBind, which has been pretrained on large-scale image–text pairs. Several recent studies have demonstrated that fine-tuning such models can yield high detection accuracy for synthetic images and can learn to discriminate model-based artifacts—even without task-specific training [4, 5].  Consequently, after fine-tuning with SBI image–text pairs, our LVLM is able to detect not only blending artifacts but also model-based artifacts.
> > >
> > > [4] Ojha, U., Li, Y., & Lee, Y. J. (2023). Towards universal fake image detectors that generalize across generative models. CVPR, 24480-24489.
> > >
> > > [5] Khan, S. A., & Dang-Nguyen, D. T. (2024). Clipping the deception: Adapting vision-language models for universal deepfake detection. ICMR 2024, 1006-1015.

---

### Official Review · Reviewer_KySN · 2025-03-18

**Overall Recommendation:** 2

**Summary:**

This papers introduces a method based on large vision language models (LVLMs) for the task of deepfake detection. To this end, the authors proposed a number of modules to enhance LVLMs performance on deepfake detection, including a knowledge-guided forgery adaptation module (KFD), a multi-modal prompt tuning framework and an iterative refinement strategy. As for data, the authors prepared their multimodal training data based on the real videos in FF++. In experiments, the authors conducted comprehensive experiments, including intra-dataset evaluation, cross-dataset evaluation, cross-manipulation evaluation of KFD, GradCAM visualization, etc. Experimental results demonstrate the effectiveness of the proposed method.

## update after rebuttal
The authors have partially addressed my concerns in the rebuttal phase. However, based on the authors' responses, I believe that a significant portion of the main paper content requires substantial revision, such as the missing demonstrations for multi-turn dialogue capabilities and details for metric calculations. I feel uncertain whether such changes can be properly reflected in the revised version of the paper. Therefore,  I will maintain my original rating and recommend that the authors consider resubmitting the revised paper to a future conference.

**Claims And Evidence:**

Some of the claims are not fully supported by experimental results.
For example, the authors claimed that their method not only supports deepfake detection but also facilitates multi-turn dialogues in Section 4.3.
However, the results in Figure 5 seems only cover the results for single-turn dialogues.

**Essential References Not Discussed:**

The authors have conducted comprehensive literature review.

**Experimental Designs Or Analyses:**

Most of the experiments are sound and valid.
However, I have some concerns over the results in Cross-Manipulation Evaluation of KFD, namely Table 2.
Based on my understanding, the training of CADDM typically required fake data. How to the authors trained CADDM with only real data in this section?

**Methods And Evaluation Criteria:**

In this paper, the authors mainly used the criteria of video-level Area Under the Receiver Operating Characteristic Curve (AUC), namely video-level AUC.
Based on my understanding, the video-level AUC requires a probability score for each video, and such a probability score is usually calculated by averaging the probability score of sampled frames in each video.
However, in this paper, the authors' method can only provide 'yes' or 'no' for each sampled frame, namely a one-hot vector instead of a probability score. How do the authors calculate the video-level AUC then, by considering the fraction of 'yes' responses in all responses of sampled frames?
If so:
- How do the authors extract 'yes' from models' responses, with chatgpt or manual rules?
- Is it still fair to compare with other SOTA-methods, where they calculate video-level AUC based the mean probability score of sampled frames?
- How many sampled frames are used per video?

I believe such discussions are required.

**Other Comments Or Suggestions:**

None.

**Other Strengths And Weaknesses:**

- One major weakness of the proposed method is its inference time.
Using the Vicuna-7B model for inference could significantly slow down the inference speed compared with other SOTA methods, which typically contain less than 1B parameters.
It is recommended that authors should discuss about the inference time and the overall throughput.
- I did not find the specific details for "An iterative refinement strategy enabling multi-turn dialog for evidence-based reasoning", which is mentioned in the abstract. I feel like this paper is still unfinished.

**Questions For Authors:**

None.

**Relation To Broader Scientific Literature:**

The task of this paper is to perform generalized deepfake detection based on LVLMs.
Firstly, this could expand the applicability of LVLMs to the area of deepfake detection, further fostering the development of AGI.
Besides, this could also boost the development of deepfake detection, by presenting models of better performance.

**Theoretical Claims:**

N/A

---

> ### Author Rebuttal · Authors · 2025-04-01
>
> **1) Multi-turn Dialogue Capabilities:** We appreciate your feedback. Following the strategy in AnomalyGPT, the alternating training strategy (Section 3.3, Implementation Details) inherently preserves Vicuna-7B’s multi-turn dialogue capabilities. While Figure 5 illustrates single-turn examples for clarity, the supplementary material (available on https://anonymous.4open.science/r/DFDGPT-8E5C) includes multi-turn dialogue screenshots. We apologize for the submission negligence and confirm full accessibility of all examples post-publication.
>
> **2) Video-level AUC Calculation for LLM:** The video-level AUC is computed by aggregating frame-level binary outputs. For each video, we sample 32 frames uniformly and calculate the ratio of “yes” responses (indicating “fake”). This ratio indicates the video’s probability score to be a fake. To extract “yes” response from model output, we implemented a deterministic rule-based parsing strategy for extracting binary labels ("yes"/"no") from model outputs. If the output contains "yes" or "is deepfake", the frame is labeled fake. If the response contains "no" or "not deepfake", the frame is labeled real. If none of the keywords are detected, the response is default labeled real. It is the common strategy to SOTA LLM-based methods and we use the same sampling strategy (32 frames/video) and aggregation (mean pooling) for fair comparison. We will add this clarification in Section 4.1.
>
> **3) CADDM Data Simulation:** Thank you for identifying this mistake. Yes, CADDM used fake data for training. We corrected it in Table 2.
>
> **4) Inference Time:** We add the evaluation of inference time on the CDF2 dataset. The inference time in our method is mainly consumed by the LLM we used, as shown in Table A. Our method with VLM model only takes more time for inference than CADDM, but get much better precision. Our method incorporating both the VLM and LLM takes more time than FAK-Owl, but also have much better precision. In addition, FAK-Owl can only provide the binary (Yes/No) responses and lacks multi-turn dialogue capabilities. As a whole, despite taking more inference time, our method has better precision and enables explainability and generalization.
>
> ||Inference time per frame(s)|AUC|
> | :-: | :-: | :-: |
> |CADDM|0\.026|85\.68|
> |Ours-VLM|0\.059|97\.62|
> |FAK-Owl|0\.642|69\.84|
> |Ours-LLM|1\.211|95\.97|
>
> **5) Iterative Refinement Strategy:** The iterative refinement strategy refers to the alternating training between the deepfake detection task and general visual dialogue task, as described in Section 4.1 (Implementation Details). This strategy enables our model to retain multi-turn reasoning capabilities by optimizing forgery detection loss and dialogue loss cyclically (see the supplementary materials in https://anonymous.4open.science/r/DFDGPT-8E5C).

---

### Decision · Program_Chairs · 2025-05-01

**Decision:**

Accept (poster)

**Comment:**

This paper was reviewed by 3 experts in the field who provided detailed suggestions. The reviewers’ recommendations were divergent, with reviewer MUQX recommending Weak Accept, and reviewers KySN and Led7 recommending Weak Reject.

All three reviewers noted the paper's strengths in addressing the important problem of generalizable and explainable deepfake detection by leveraging VLMs. Their concerns, however, included insufficient validation and detail for the claimed multi-turn dialogue capabilities (KySN, Led7), computational cost and inference time due to the large models used (KySN), the lack of quantitative metrics for evaluating the claimed localization and explanation quality (MUQX, Led7), and insufficient comparison with the most recent state-of-the-art methods (MUQX). Methodological clarity regarding AUC calculation from binary outputs and the representativeness of the simulated training data were also questioned (all three reviewers).

I read the reviews and rebuttal and recognize the potential merits in the paper. Though I share the reviewers' concerns, these seem to focus more on how the method was evaluated (and where the results were to be found in the supplemental) rather than the essence of the proposed contribution. These concerns, while valid, do not rise in my opinion to the level of a reject recommendation and so recommend accordingly.